# Inferring interiors and structural history of top-shaped asteroids from external properties of asteroid (101955) Bennu

Yun Zhang [1,2] ✉, Patrick Michel [1], Olivier S. Barnouin [3], James H. Roberts [3], Michael G. Daly [4], Ronald-L. Ballouz [3,5], Kevin J. Walsh [6], Derek C. Richardson [7], Christine M. Hartzell [2] & Dante S. Lauretta [5]

Asteroid interiors play a key role in our understanding of asteroid formation and evolution. As no direct interior probing has been done yet, characterisation of asteroids' interiors relies on interpretations of external properties. Here we show, by numerical simulations, that the top-shaped rubble-pile asteroid (101955) Bennu's geophysical response to spinup is highly sensitive to its material strength. This allows us to infer Bennu's interior properties and provide general implications for top-shaped rubble piles' structural evolution. We find that low-cohesion ($\lesssim 0.78$ Pa at surface and $\lesssim 1.3$ Pa inside) and low-friction (friction angle $\lesssim 35°$) structures with several high-cohesion internal zones can consistently account for all the known geophysical characteristics of Bennu and explain the absence of moons. Furthermore, we reveal the underlying mechanisms that lead to different failure behaviours and identify the reconfiguration pathways of top-shaped asteroids as functions of their structural properties that either facilitate or prevent the formation of moons.

Rubble-pile structures are expected to predominate among asteroids with diameters of 200 m to tens of kilometers according to the current understanding of asteroid formation and collisional evolution[1]. Observations indicate that a large fraction of these asteroids resemble a spinning top, i.e., an oblate spheroid with an equatorial bulge rotating about its short axis, with many being accompanied by moons[2]. Given that rubble-pile structures are susceptible to reconfiguration by changes in spin states, the formation of top-shaped asteroids and binary systems is commonly attributed to spinup via the thermal Yarkovsky-O'Keefe-Radzievskii-Paddack (YORP) effect[3–5]. However, the geophysical expressions and evolution of top-shaped asteroids driven by this effect and their dependency on asteroid interior properties are poorly understood[1].

To date, little is known about the internal structure and material properties of asteroids except for some indirect indications regarding the internal density distribution[6,7]. Previous studies showed that the reshaping process of a spherical rubble-pile body under YORP spinup would be altered by the amount and distribution of material properties within this body[8–11] but lacked connections with actual asteroid surface data. The next challenge is to use detailed geophysical features returned by spacecraft equipped with state-of-the-art instruments to constrain or even identify the interior properties as well as the associated structural history of visited asteroids. This provides an opportunity to infer evolutionary scenarios from these asteroids' origin to their current states along with important insights for future asteroid space exploration. In this study, we tackle this challenge.

Three top-shaped asteroids have been visited by spacecraft, i.e., (2867) Steins by the ESA's Rosetta mission[12], (162173) Ryugu by the JAXA's Hayabusa2 mission[13], and (101955) Bennu by NASA's Origins, Spectral Interpretation, Resource Identification, and Security-Regolith Explorer (OSIRIS-REx) mission[14]. With dedicated rendezvous investigations, the latter two missions revealed tremendous details on the

[1]Université Côte d'Azur, Observatoire de la Côte d'Azur, CNRS, Laboratoire Lagrange, Nice, France. [2]Department of Aerospace Engineering, University of Maryland, College Park, MD, USA. [3]The Johns Hopkins University, Applied Physics Laboratory, Laurel, MD, USA. [4]The Centre for Research in Earth and Space Science, York University, Toronto, ON, Canada. [5]Lunar and Planetary Laboratory, University of Arizona, Tucson, AZ, USA. [6]Southwest Research Institute, Boulder, CO, USA. [7]Department of Astronomy, University of Maryland, College Park, MD, USA. ✉e-mail: yun.zhang@oca.eu

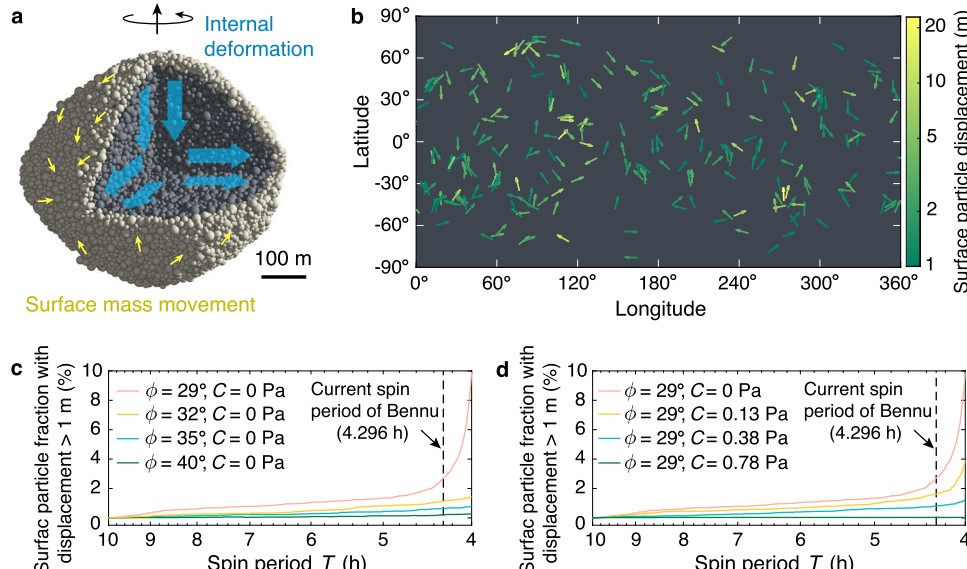

**Fig. 1 | Surface mass movement on the Bennu-shaped rubble pile during YORP spinup. a** Cut-away schematic of the Bennu-shaped rubble-pile model, where surface particles are highlighted in beige (i.e., the top 25-m-depth layer, where the total particle number is 11,148). The blue and yellow arrows indicate the general internal deformation and surface mass movement direction. The rotation direction is indicated by the black arrows on the top. **b** Surface particle movement map in the case of friction angle $\phi = 29°$, cohesion $C = 0$ Pa. The displacement measures the distance between a surface particle's initial position and its position at $T = 4.296$ h in the body-fixed frame. **c, d** The percentage of surface particles with displacement larger than 1 m for simulations with different friction angles and cohesive strengths, respectively. Source data are provided as a Source Data file.

diverse surface geophysical features of the explored targets. Taking the carbonaceous asteroid Bennu as an example, images obtained by the OSIRIS-REx spacecraft confirmed that Bennu is a 500-m-diameter rubble pile with a top-like shape[15,16]. There are several large craters approximately 100 m in diameter distributed on Bennu's equator[17]. These craters are stratigraphically younger than the equatorial bulge and may require up to 65 Myr Bennu's residence time in the main belt to form[18]. Driven by the YORP torque, Bennu is currently experiencing a rotational acceleration of $3.63 \pm 0.52 \times 10^{-6}$ degrees day$^{-2}$ (refs. [19,20]). Assuming that the current spinup is steady, the time required for Bennu's spin rate to double is about 1.5 Myr in its current near-Earth orbit and about 6 Myr in the inner main belt. These timescales are considerably smaller than the surface age of the equatorial bulge, raising questions on how to preserve these large craters during rotational instability[21,22]. Some local mass movements in the downslope direction are also detected[23]. This direction and the associated colour variation analyses suggest that these mass movements occurred within the past 0.2 Myr, while the spin rate of Bennu was already close to the current rate[23,24]. There are also some long linear features on the surface, which may suggest internal stiffness[16]. In addition, according to the measurements of its gravity field, Bennu has a heterogeneous mass distribution and is likely to have a lower-density interior[7]. Interpretation of this complex set of information require a comprehensive understanding of asteroid geophysical processes and their proper modelling.

Here we perform numerical simulations using the Soft-Sphere Discrete Element Method (SSDEM)[25,26] to test Bennu's structural response to YORP-induced spinup. By exploiting the numerical results and making comparisons with Bennu's geophysical features, we quantify the interior properties and derive a unified evolutionary scenario for Bennu, which allows us to draw general implications for the structural evolution of top-shaped rubble piles.

## Results
To capture the small-boulder distributions and shape of Bennu at high resolution, we model Bennu as a self-gravitating rubble pile consisting of approximately 41,000 spherical particles with radii ranging from

about 4 to 12 m following a power-law size distribution (Fig. 1a; see Methods Section Bennu rubble-pile model). The SSDEM N-body code PKDGRAV with quantitatively controllable material strengths is applied to integrate the motion of particles (see Methods Section Soft-sphere discrete element method). A quasi-static spinup procedure is designed to model the YORP spinup process (see Methods Section Modelling of YORP-induced quasi-static spinup). To explore a wide range of possible geological material properties, numerical experiments are run with four friction angles $\phi$ (i.e., 29°, 32°, 35°, and 40°) and a series of macroscopic cohesion values $C \in [0, 22]$ Pa. We first consider homogeneous structures with uniform material strengths to investigate material properties' influence on the failure behaviours, and then use layered structures, i.e., consisting of cohesive interiors and cohesionless surfaces, and heterogeneous structures, i.e., consisting of local high-cohesion internal regions and cohesionless matrix, to infer Bennu's structural properties.

### Typical failure behaviours of top-shaped rubble piles
Based on our spinup numerical experimental results, four distinct rotational failure modes are identified for the Bennu rubble-pile model, i.e., Type I local surface landslides, Type II internal deformation, Type III mass shedding, and Type IV tensile disruption (Supplementary Table 1). Among these failure modes, the Type I mode only causes regional surface mass movement in some high slope regions and the rubble pile can be continuously spun up and keep its overall structure stable at faster spin; conversely, Type II–IV modes induce a spin-down effect due to either global reconfiguration or mass ejection and faster spin states are therefore prohibited. Supplementary Movies 1–12 present the surface and interior structural evolution of several typical cases, showing that a rubble-pile model can exhibit multiple failure behaviours during spinup. By computing the dynamical surface and internal slope/cohesion of the rubble pile at failure (see Methods Section Dynamical internal/surface slope and cohesion distribution of a rubble pile; Eqs. (5)–(10)), we find that the surface and internal response of the Bennu-like rubble pile to YORP spinup is highly sensitive to the material properties, which determine the failure mode.

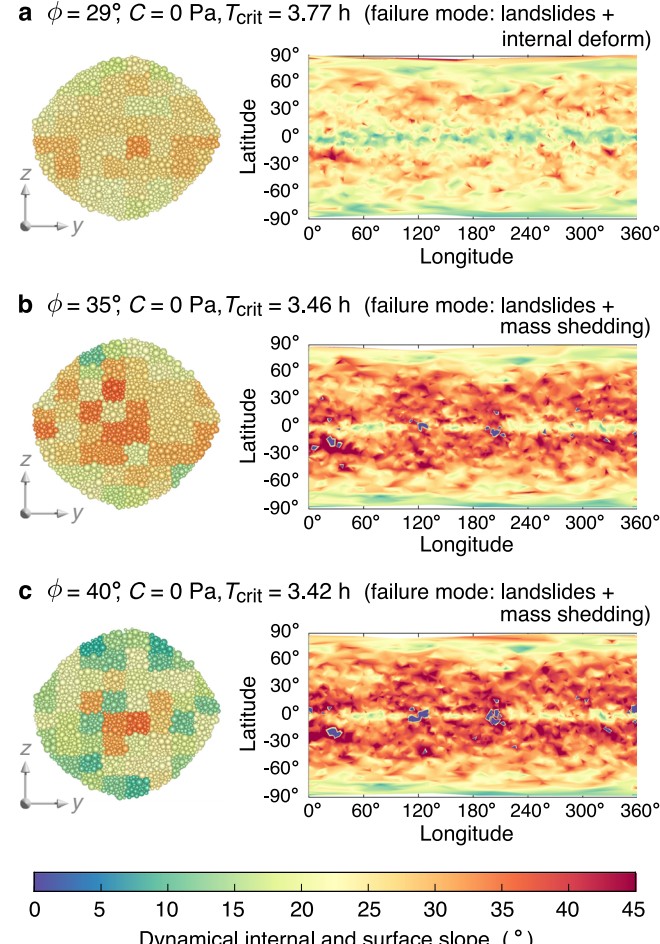

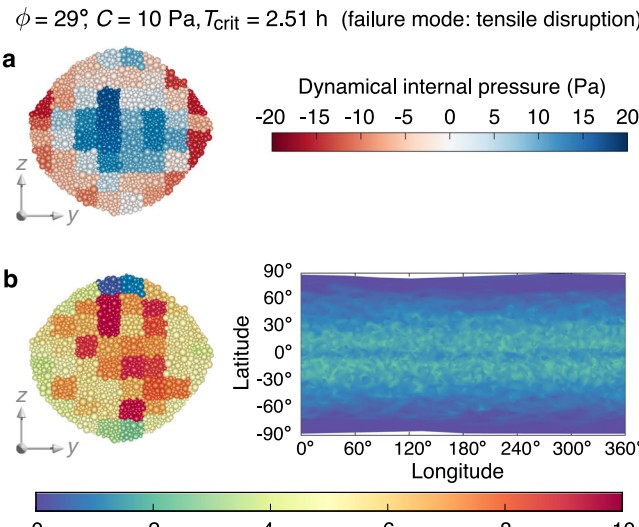

**Fig. 3 | Pattern of internal and surface failure regions of a cohesive rubble pile.** Dynamical internal pressure (**a**; tensile stress is expressed as a negative value) and internal and surface cohesion map (**b**) at the critical spin limit $T_{crit}$ for a simulation with strong cohesion ($C = 10$ Pa). Regional failure is initiated when the local dynamical internal/surface cohesion exceeds the value of the cohesive strength $C$, which determines the failure behaviours as indicated on the top of this figure. The dynamical internal pressure and cohesion are calculated based on the averaged-stress analysis using representative volume elements (the patches shown represent the elements in these cross-sections), and the dynamical surface cohesion is derived based on the rubble-pile alpha-shape model (Methods). Source data are provided as a Source Data file.

**Fig. 2 | Pattern of internal and surface failure regions of cohesionless rubble piles. a–c** Dynamical internal slopes over a cross-section and surface slope map at the critical spin period $T_{crit}$ for simulations with three friction angles $\phi$, respectively. The surface regions where material can be lofted are marked with negative slopes (the bluish areas). Regional failure is initiated when the local dynamical internal/surface slope exceeds the value of $\phi$, which determines the failure behaviours as indicated on the top of each panel. The dynamical internal slope are calculated based on the averaged-stress analysis using representative volume elements (the patches shown represent the elements in these cross-sections), and the dynamical surface slope are derived based on the rubble-pile alpha-shape model (Methods). Source data are provided as a Source Data file.

Taking the homogeneous Bennu-shaped model as an example, when the material is cohesionless, the failure mode gradually switches from Types I & II to Types I & III with increasing friction angle (Supplementary Movies 1, 2, 4–6). For instance, when $\phi = 29°$, both the surface and interior locally violate the stability criterion at the spin limit (i.e., surface/internal slopes ≥ $\phi$), leading to landslides and internal deformation (Figs. 1b, c and 2a). The body is reconfigured into a more oblate shape with slower rotation, by conservation of angular momentum, and a less dense interior due to shear dilatancy (Supplementary Fig. 1). For $\phi \gtrsim 35°$, some local surface regions near the equator experience outward accelerations before internal slopes exceed $\phi$ (Fig. 2b, c). Particles at these regions can be lofted into some close orbits above the surface, and the spin rate of the body slightly decreases to conserve angular momentum in the system.

When the material is cohesive, surface landslides are completely inhibited when $C \gtrsim 0.78$ Pa (Fig. 1d), and the failure mode gradually evolves from Type II to Type IV with increasing cohesion (Supplementary Movies 3, 7). As shown in Fig. 3b, the required cohesion for surface stability is much smaller than that for internal stability. Therefore, cohesive rubble piles are prone to fail internally. When $C \gtrsim 10$ Pa, surface layers are subject to intensive tensile stress as revealed by the dynamical internal pressure distribution in Fig. 3a. In this case, when the interior starts to deform, surface fractures are generated along the longitudinal directions, leading to tensile disruption.

## A unified phase diagram for failure-mode diagnosis

According to the discovered instability patterns and stress/slope distribution characteristics of each failure mode, we derived theoretical failure conditions (see Methods Section Theoretical failure conditions) and constructed a unified phase diagram (Fig. 4) that can quantitatively predict the structural evolution of a Bennu-shaped rubble pile. The same semi-analytical approach could be applied to other bodies with a different shape for assessing their structural properties and evolution without running any SSDEM simulations.

The limiting spin period that can activate each failure mode in a rubble pile with given material properties is plotted in Fig. 4. Since the failure condition is evaluated based on the stress state at different failure locations for different failure modes and the stress state in a rubble pile varies significantly within the body (see Fig. 2 for examples), the limiting spin periods of the body for different failure modes are substantially different from each other. During a spinup process (from left to right in Fig. 4a, b), the body will exhibit the corresponding failure behaviour when its spin period decreases to the limiting spin period of a certain failure mode, i.e., by reaching the corresponding failure curve. As landslides only occur locally at some high slope regions and the slopes at these regions can be reduced concurrently with the landslides[27], the body can enter the right-hand sides of the Type I curves in Fig. 4 when it is spun up by the YORP effect. However, as Types II–IV involve large reconfiguration/mass ejection and can trigger substantial spin-down, spin

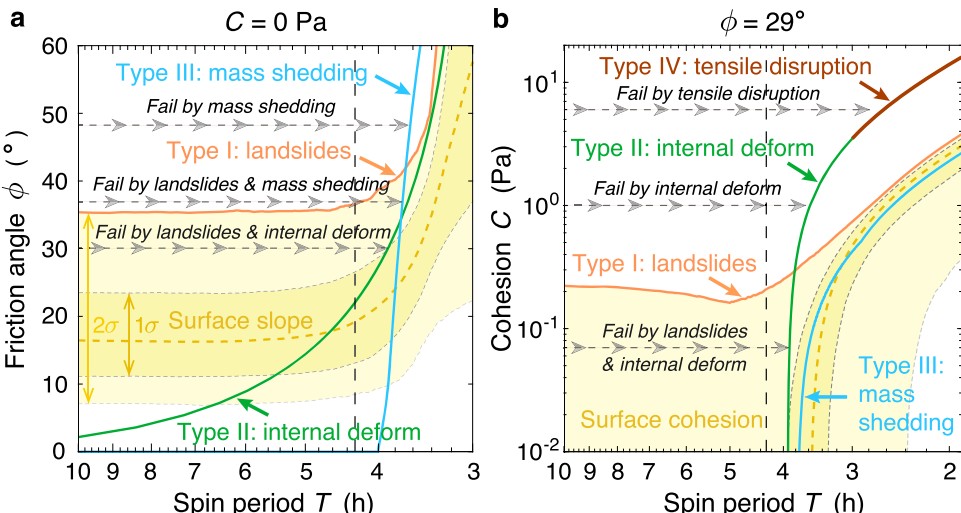

**Fig. 4 | Bennu-shaped rubble-pile failure-mode diagram.** Failure spin periods of the four failure types as functions of the material friction (**a**, where $C = 0$ Pa) and cohesion (**b**, where $\phi = 29°$) are shown by the solid curves with different colours (see Methods Section Theoretical failure conditions for the procedure to generate this diagram). The grey horizontal dashed lines with arrowheads represent some possible evolutionary pathways of rubble piles during YORP-induced spinup. Depending on the material properties, a rubble pile would end up reaching different failure curves, where its structure would fail via the corresponding failure types as indicated by the grey text. The distributions of surface slopes and cohesion of the Bennu-shaped rubble pile are shown as the median (i.e., the dashed yellow curves) and $1\sigma$ to $2\sigma$ ranges (i.e., the yellowish regions with different opacity as indicated by the double-sided arrows), indicating that only a small portion of surface area is subject to high slope/cohesion. Therefore, rubble piles can cross the Type I failure curves with local landslides, but spin periods shorter than the Types II–IV failure curves are unreachable because of global structural failure. At Bennu's current spin period (as indicated by the black vertical dashed line), surface landslides could be initiated when the material friction and cohesion are small, and its internal structure could deform if its material is cohesionless and friction angle is smaller than 22°. Source data are provided as a Source Data file.

states in the right-hand sides of the Types II–IV curves are strictly forbidden. Therefore, for a given $\phi$ or $C$, the failure mode can be deduced from this diagram.

The horizontal dashed lines with arrowheads in Fig. 4 illustrate the spinup process and the failure mode of a rubble pile with given material properties. Without cohesion (Fig. 4a), when $\phi = 30°$, the body would first enter the Type I failure region during spinup and experience local landslides at slow spin and then cross the Type II curve and be globally reshaped through internal deformation; when $\phi \gtrsim 35°$, local landslides could still occur, but the body would eventually cross the Type III curve and shed mass to offset the rotational acceleration at fast spin. When cohesion is present (Fig. 4b), the Type III curve is unreachable in the case of $\phi = 29°$ as the body would first cross the Type II curve. Similarly, the body can never go through the Type I failure when $C \gtrsim 0.3$ Pa ($2\sigma$).

The dependency of failure modes on asteroids' material properties revealed in the current study is much more complex and sensitive than previous studies predicted. It has been suggested, based on both continuum-theory analyses[8] and SSDEM simulations[9], that in homogeneous spherical rubble piles, the internal core would always fail before the surface shell. However, our results show that a rubble-pile model with homogeneous structures can also fail superficially via the Type III mass-shedding mode when the friction angle is sufficiently large. The internal failure mode (i.e., the Type II failure mode) only occurs for some combinations of friction and cohesion values. This discrepancy could be mainly due to the utilisation of increased-resolution SSDEM models and the quasi-static spinup procedure in the current study that can characterise the geophysical features (such as the shape and surface slopes) and YORP-induced structural response of an actual asteroid. Both cannot be captured by the static continuum analyses and the lower-resolution SSDEM models used in these previous studies. This high sensitivity of the failure mode to the material properties revealed here lays out the foundation for interpreting the interiors and structural histories of a rubble-pile asteroid such as Bennu.

## Structural properties and evolution of Bennu

Given that morphological evidence for surface movement of boulders and smaller particles has been found at most locations on Bennu[23], our results indicate that Bennu is unlikely to have large surface cohesion globally. The downslope mass movement signatures and surface flow distribution on Bennu[23] match well with our simulation results using small or zero cohesion (Fig. 1b). According to the surface mobility analyses shown in Fig. 1d, Bennu's surface cohesion should be $\lesssim 0.78$ Pa. This minimal surface cohesion characteristic can account for the crater morphology[28] and the terrace expressions[27] observed on Bennu, and is also consistent with the fact that small resistance forces were experienced during the penetration of the sampling head of the OSIRIS-REx spacecraft[29,30].

Nonetheless, Bennu could still have a highly cohesive interior. A few craters on Bennu have central mounds, implying the presence of material with increased strength located a few metres beneath the surface[31]. Some surface lineament features are also indicative of internal cohesion[16]. To determine whether this possibility could actually represent Bennu, we performed simulations of spinup evolution of layered structures with a cohesionless surface and a cohesive interior.

Due to the high surface mobility of these layered structures, Type I landslides are ubiquitous during the spinup processes (Supplementary Movies 8–10). These rubble piles would fail by Type II internal deformation ($C < 1.3$ Pa) or Type III mass shedding ($C \gtrsim 1.3$ Pa) at rapid spin. The failure-mode transition between Types II and III happens when the interior cohesion is sufficiently strong to resist the internal shear stress. This can be inferred from the failure-mode diagram. For example, based on Fig. 4a, the critical spin limit for mass shedding is about 3.7 h for a cohesionless surface with $\phi = 29°$. By imposing this spin limit on Fig. 4b, the corresponding critical cohesion for internal deformation is about 1 Pa.

Below this critical cohesion, the interior of the body first crosses the Type II curve of Fig. 4b and would deform before being able to shed mass from the surface. This failure mode through internal deformation naturally prevents YORP-induced satellite formation[4] and would

preserve the large equatorial crater topography as observed on Bennu. Above this critical cohesion, Type III mass shedding occurs. The orbits of ejected particles would be within the region where they could efficiently accumulate into a satellite[32] (Supplementary Fig. 2). In effect, the rubble-pile structures of both Bennu and a close-in accumulated satellite could help to reach a long-term stable equilibrium for a synchronous system[33]. However, currently no natural satellite is orbiting Bennu, and the detected ejection of small grains from Bennu's surface is more likely due to other mechanisms (e.g., thermal fracturing, dehydration, and/or meteoroid impacts) rather than the YORP spinup[34]. Furthermore, the substantial mass shedding and possible subsequent mass deposit induced by this Type III failure would not allow preserving the large equatorial crater topography.

The better fit of Type II over Type III puts strong constraints on both the global friction angle and the overall internal cohesion of Bennu. First, the global friction angle must be <35°. This threshold is lower than the maximum slopes found on Bennu's surface[16], which extend to more than 60°. These steep local surface regions are on metre-length scales and could be evidence of cohesion[16,27]. To infer the material strength for sustaining Bennu's surface slopes, we carried out avalanche simulations under Bennu's microgravity (see Methods Section Avalanche numerical experiments). The results show that a tiny amount of cohesion $C = 0.038$ Pa can keep a 2-m-thick layer with $\phi = 29°$ stable at slopes $\geq 60°$. The required cohesion is much smaller than the surface-boulder-movement threshold of 0.78 Pa. Therefore, a relatively low-friction surface with $C$ about $0.01–0.1$ Pa can consistently account for both the detected surface topography and mass movement activities on Bennu.

Second, the overall internal cohesion must be <1.3 Pa. This threshold is substantially smaller than the local subsurface cohesion (140–670 Pa) deduced from the outcome of the artificial impact experiment on the asteroid Ryugu by the Hayabusa2 mission[35]. Considering that the main source of cohesion in airless bodies is thought to arise from interparticle van der Waals forces, which are linked to particle size and surface properties[36], the remarkable heterogeneity in particle sizes[17,37] and compositions[38] detected on Bennu would imply a nonuniform internal cohesion distribution[22].

To explore the evolution of heterogeneous rubble piles, we constructed models with local strong cohesive regions embedded in a cohesionless matrix, and performed spinup simulations. To draw more general implications for top-shaped rubble piles and eliminate the effect of shape asymmetry, we used the longitudinally averaged profile of Bennu's northern hemisphere to shape these models (see Methods Section Bennu rubble-pile model). The results show that internal and surface particle flows are profoundly influenced by the presence of these strong regions. The failure mode is Types I & II when $\phi < 35°$ and Types I & III when $\phi \gtrsim 35°$ (Supplementary Movies 11–12).

Figure 5 presents the results of the heterogeneous model with four strong regions near the equator and one in the southern hemisphere, with $\phi = 29°$, and whose reconfiguration process can systematically explain all Bennu's known geophysical features: (1) the Types I & II failure mode is consistent with the recent surface mass movement[23], the absence of moons, and the large-crater retention in the equatorial bulge[17]; (2) the heterogeneous landslides facilitate the formation of a non-circular equator (Fig. 5e), and the accompanying internal deformation, which is symmetric along the $x = 0$ and $y = 0$ axis (Fig. 5c,d), leads to a squarish equatorial shape (Fig. 5b), resembling the shape of Bennu[16]; (3) the cohesive region in the southern hemisphere is pushed towards the centre and surface particle flow is less intense in the southern hemisphere (Fig. 5c, e), resembling the north-south hemispherical differences found on Bennu, where the north has more evidence of boulder movement and surface flow[23,37]; (4) surface flow near the equator could be at the origin of the mass wasting signatures observed on Bennu's equator[16]; and (5) the outward movement of the strong equatorial regions and internal shear deformation result

in a large decrease in the internal filling efficiency and a dilute core (Supplementary Fig. 3), consistent with Bennu's gravity field measurements[7].

## Discussion

By linking observed geophysical features with numerical experiments, we probe Bennu's interior and identify associated paths in its structural evolution. We find that surface cohesion $\gtrsim 0.78$ Pa would impede any detectable surface boulder movement, and homogeneous interior cohesion $\gtrsim 1.3$ Pa would cause mass shedding and facilitate moon formation; both scenarios are contrary to Bennu's observed features. Instead, low-cohesion and relatively low-friction structures with several local high-cohesion internal regions could systematically account for all the known geophysical characteristics of Bennu.

The inferred globally low-strength interior supports the idea that Bennu was formed by gravitational reaccumulation of fragments following the catastrophic collision of its parent body[39]. Strong cohesive local internal zones could be due to the presence of large boulders or local concentrations of cohesive fine grains[40] that existed during the reaccumulation or were produced later in Bennu's evolution. As spherical rubble piles formed via reaccumulation generally have low resistance to shear stress[39,41], our best-fit structural and material properties can consistently link Bennu's current features and evolutionary history with its collisional origin.

The low surface cohesion may be due to the lack of dust and sub-mm-sized grains on Bennu's surface[40,42]. However, the reason why fine grains are not present remains a question, and may be related to the efficiency of processes, such as impact comminution and thermal fatigue, for grinding down carbonaceous materials to fines[43]. Some mechanisms, such as electrostatic forces[44], could also loft and remove fines from asteroid surfaces. Another possibility is that fines percolate into the subsurface thanks to the large structural macroporosity and/or seismic shaking effects[45,46], which may explain the presence of subsurface sub-mm-sized grains observed during the OSIRIS-REx mission's sampling operation[47]. Nonetheless, our prediction of the regional lack of cohesion in Bennu's interior implies that fine sedimentation may not be effective globally in such a small body.

Our analysis of Bennu allows inferring interior properties from observed surface characteristics and goes beyond Bennu itself by identifying the different structural evolutionary paths of top-shaped asteroids. In particular, if carbonaceous rubble piles have similar material properties as those inferred for Bennu, the underlying low-strength structures lead to internal deformation in response to YORP spinup and consequently to the inhibition of satellite formation. Thus, we predict that the lack of both fines and moon generation by spinup processes could be characteristics of asteroids of carbonaceous type in general[48].

## Methods

### Bennu rubble-pile model

Bennu is explicitly modelled as a self-gravitating rubble pile in our numerical investigation. A two-step procedure is designed to construct the Bennu-shaped rubble-pile model and can be applied to any kind of rubble pile. First, a Bennu-sized granular assembly is created by simulating the gravitational collapse of a spherical cloud of particles with a predefined size distribution. To mimic a rubble-pile configuration experiencing impact induced global seismicity[46], particles in the assembly are assigned with random velocities within the range of 0 to 5 cm/s, and the assembly is then allowed to settle down under its own self-gravity. Frictionless material parameters are used for the above procedure to obtain a macroscopically homogeneous close-packing aggregate[45]. Next, we use the Bennu shape model derived from the data collected by the OSIRIS-REx Laser Altimeter[37,49] (OLA v20) to shave extra particles off the assembly to resemble Bennu's shape, and settle down this Bennu-shaped model at a spin period of $T = 10$ h with

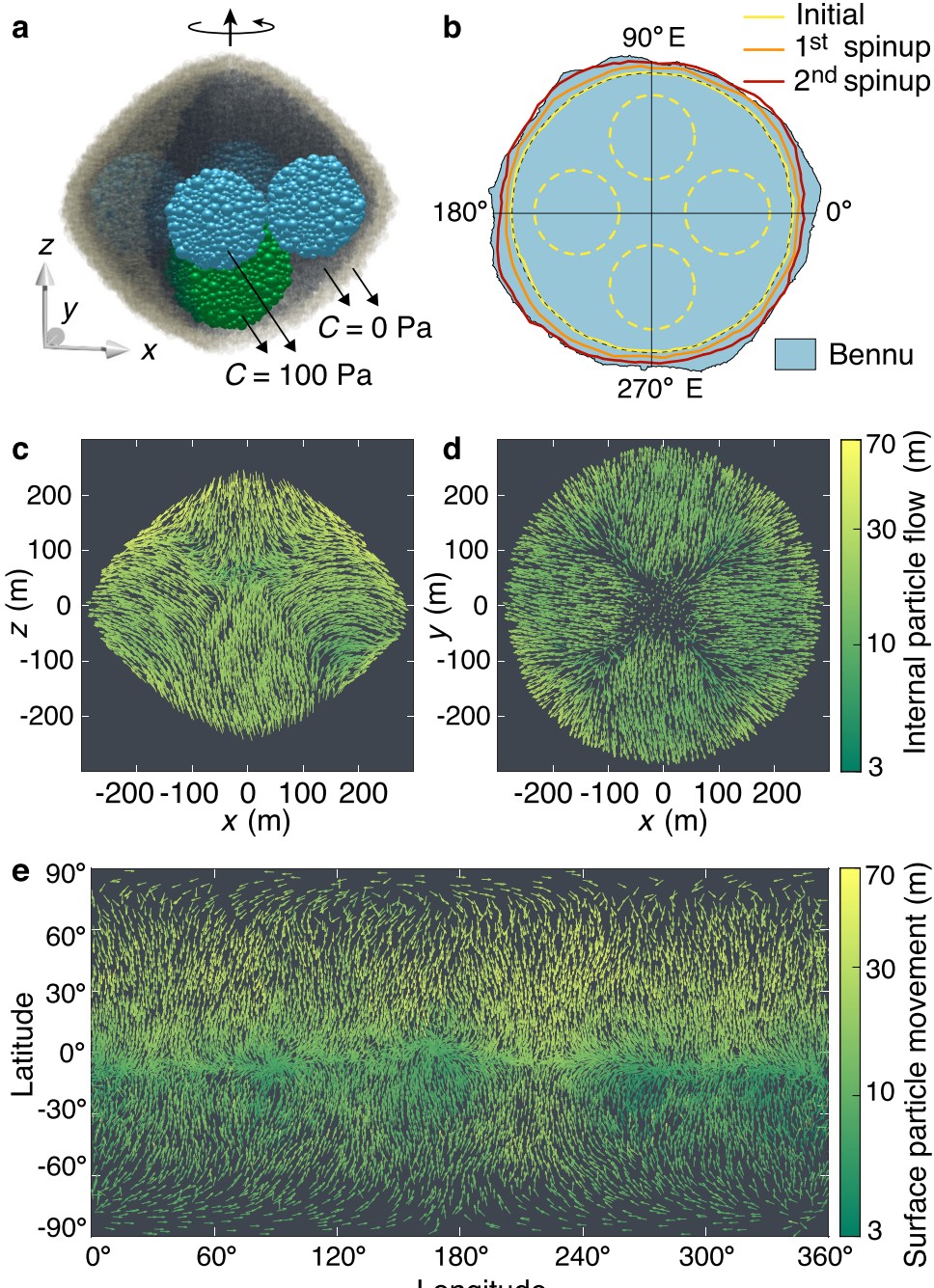

**Fig. 5 | Reconfiguration and resurfacing of a heterogeneous top-shaped rubble pile. a** Cut-away translucent schematic of the heterogeneous model, where surface particles are highlighted in beige and internal strong cohesive spherical regions ($C = 100$ Pa) are highlighted in blue (whose mass centres located at the equatorial plane and radii = 80 m) and green (whose mass centre located at $(x, y, z) = (0, 0, -100)$ m and radius = 100 m). The friction angle $\phi = 29°$. Two consecutive spinup-settling paths are applied to this model for testing its structural evolution (Supplementary Fig. 3). **b** Equatorial profile of the rubble pile at the beginning (near spherical) and the end of the two spinup-settling paths. The initial positions of the four equatorial strong regions are indicated by the yellow-dashed circles. **c, d** Internal particle flow of the first spinup-settling path over a centre cross-section parallel to the $x-z$ and $x-y$ planes, respectively. **e** Surface particle movement map of the first spinup-settling path. Source data are provided as a Source Data file.

the corresponding material friction parameters for different tests. Supplementary Fig. 4 compares the derived Bennu-shaped rubble-pile model with the OLA shape model, showing good agreements.

The constructed Bennu-shaped rubble-pile model consists of approximately 41,000 spheres with radii ranging from about 4 to 12 m, which represent the small-boulder constituents of Bennu. According to the size-frequency distribution of boulders with similar radii on Bennu[15,17], the particle size distribution is set to follow a differential power law with an exponent of −3. Due to the limit of

current computational power, smaller cobbles, pebbles, and fine grains are not explicitly simulated, but their cumulative friction and cohesion effects on the metre-sized boulders are captured by our discrete element modelling[10]. As demonstrated in our previous spinup simulations of top-shaped rubble piles[50], the failure behaviour and condition are not sensitive to the particle size resolution. Therefore, this rubble-pile model provides a reliable and computationally feasible numerical representation of Bennu for the aim of this study.

The internal packing efficiency of this rubble-pile model, i.e., the fraction of the volume in the model's internal structure that is occupied by the constituent particles (which is evaluated based on the Voronoi tesselation[51]), $\eta_{inter} \approx 67\%$, is consistent with the estimated range of Bennu's packing efficiency (i.e., 50–75%) inferred from its bulk density[16]. The coordination number, i.e., the average contact number of each particle, $N_{cont}$ ranges from 3.7 to 4.8, depending on the material friction and cohesion. The mass and bulk density of this model are set to $7.329 \times 10^{10}$ kg and 1190 kg/m³, respectively (the same as those of Bennu). To consider the possible mass density heterogeneity of Bennu's interior[7], we considered three density distributions: an underdense core (i.e., the mass density of the centre 100-m-radius region is smaller than that of the outer region, where $\rho_{inter} = 0.8\rho_{outer}$); a denser core ($\rho_{inter} = 1.2\rho_{outer}$); and a uniform density ($\rho_{inter} = \rho_{outer}$), respectively. Supplementary Table 1 summarises the results for the three models, which fail with the same behaviour, showing that the failure behaviour is not strongly sensitive to these models. The underdense-core Bennu-shaped rubble-pile model is taken as the representation to present the results and analyses in this study, unless otherwise specified.

For the heterogeneous rubble-pile models investigated later in this study, the northern and southern hemispheres of these models are instead carved by the longitudinally averaged profile of Bennu's northern hemisphere to eliminate the effect of shape asymmetry on structural evolution (some small scales of heterogeneous landslides and deformation have been detected in our spinup simulations with the Bennu-shaped rubble-pile model, which may obscure the effect of strength heterogeneity), and only the uniform density distribution is considered in this case. Other model properties are close to those of the Bennu-shaped rubble-pile model. Together, we refer these models as Bennu-like rubble-pile models.

Note that, due to computational constraints, previous SSDEM simulations dedicated to the study of rubble-pile failure behaviour, e.g., refs. 8,9, commonly used rubble-pile models that consisted of $N$ about 3000 spheres with size differences $R_{max}/R_{min} < 1.5$, which cannot capture the high-accuracy shape model of a space-mission target like Bennu, and thus limit the modelling of the target's geophysical features and evolution. In this study, thanks to the hierarchical tree data structure and high-efficiency parallelisation of our modelling code, PKDGRAV[52,53] (see below), a self-gravity $N$-body system with $N$ up to $10^5$ can be readily modelled within acceptable computational costs[50]. By using the increased-resolution models with relatively large particle size differences, we are able to mimic the observed boulder distribution on Bennu and capture the shape of Bennu at relatively high resolution. This increased-resolution model also enables us to design the analytical approach (see "Methods" sections Dynamical internal slope and cohesion distribution of a rubble pile, and Dynamical surface slope and cohesion distribution of a rubble pile) to calculate the internal stress-state and surface slope/cohesion distribution (see Figs. 2, 3 for examples), which helps reveal the underlying mechanisms that lead to different failure behaviour and provide insights into the surface and internal properties of top-shaped asteroids and their geophysical evolution. As demonstrated in Supplementary Fig. 5, the surface slope distribution of our rubble-pile models resembles that of Bennu, which allows us to carry out the surface mass movement analyses and make direct comparisons with the geophysical features discovered by the OSIRIS-REx mission (see Fig. 1). These modelling and data analysis techniques in turn enable us to constrain Bennu's surface and interior material properties and develop a unified evolutionary scenario for Bennu as discussed in the main text.

## Soft-sphere discrete element method

We use the high-efficiency parallel $N$-body tree code, PKDGRAV[52,53], and its SSDEM framework[26], including an improved rolling friction[51] and cohesion[10] model, to solve the gravity and contact interactions between spherical particles representing the components of a simulated rubble pile. The movement and rotation of each particle are obtained by solving the equations of motion with a second-order leapfrog integrator. Two compiled versions of PKDGRAV are provided as Supplementary Software 1.

Briefly, the SSDEM model includes a linear spring-dashpot normal contact force, a normal cohesive force, a spring-dashpot-slider tangential contact force, and two spring-dashpot-slider rotational torques in the rolling and twisting directions. A set of interparticle parameters is used to adjust the mechanical properties, including: two stiffness constants, ($k_N$, $k_S$), for controlling the compressive strength along the normal and tangential directions; two coefficients of restitution, ($\varepsilon_N$, $\varepsilon_S$), for controlling the contact energy dissipation along the normal and tangential directions; three friction coefficients, ($\mu_S$, $\mu_R$, $\mu_T$), for controlling the material shear strength along the tangential, rolling, and twisting directions; a shape factor, $\beta$, to take into account the fact that real particles are not spherical; and an interparticle tensile strength coefficient, $c$, for controlling the material cohesive strength along the normal direction. The SSDEM model and the relation between the parameter sets and mechanical properties have been calibrated with laboratory experiments on real sands[10,26]. The performance and reproducibility of this SSDEM model has been verified by comparisons with another SSDEM package, i.e., the open-source code Chrono[54].

Based on the compressive strength estimated for metre-sized boulders on Bennu[42,55], i.e., about 0.1–1 MPa, the normal stiffness $k_N$ is set to $2.0 \times 10^7$ N/m. The tangential stiffness $k_S$ is taken to be $(2/7)k_N$ to maintain the same oscillation frequencies along the normal and tangential directions[26]. The two coefficients of restitution, $\varepsilon_N$ and $\varepsilon_S$, are both taken to be 0.55, which is close to the low-speed collisional energy dissipation of terrestrial rocks[56]. $\mu_R$ and $\mu_T$ are taken to be 1.05 and 1.3, respectively, representing the rough surfaces of medium-hardness rocks[57]. To keep particle overlaps smaller than 1% of the minimum particle radius and integrate the particle contacts precisely, an integration timestep of 0.02 s is used. The free parameter set, ($\mu_S$, $\beta$, $c$), is used to adjust the material shear and cohesive strength of the simulated rubble pile. Considering the possible material heterogeneity within a rubble pile, the interaction between different particle pairs can have different values of $c$. In a homogeneous structure, a constant $c$ is used throughout the rubble-pile model; in a layered structure, $c$ is set to 0 Pa in the 25-m-depth surface layer and nonzero values in the interior; in a heterogeneous structure, only the specified cohesive regions have nonzero $c$ and particle interactions in other locations are cohesionless.

The friction angle $\phi$ and macroscopic cohesion $C$ are the two main properties that are commonly used to characterise the macroscopic strength of geological materials. By using the homogenisation and averaged-stress-analysis method introduced below, the value of $\phi$ can be derived for each interparticle parameter set. In the case where a rubble pile fails through internal deformation, we can derive its friction angle by finding the maximum internal slope according to Eq. (5) (see below) at its failure spin state, $\phi = \max_{j \in RVEs} \theta_j^{inter}|_{failure}$. In the case where it fails through surface mass shedding, $\phi$ is taken to be the angle of repose derived from our avalanche numerical experiments (see below). Although the angle of friction and the angle of repose may differ due to the different confining pressure[58], by taking the case of ($\mu_S$, $\beta$) = (0.2, 0.3) as an example, our avalanche numerical experiments show that the internal friction angle is close to the angle of repose under the microgravity environment of Bennu.

The macroscopic cohesion $C$ can then be estimated according to the packing properties of the rubble-pile model by[50]

$$C = \frac{c\beta^2 N_{cont} \eta_{inter} \tan\phi}{4\pi}. \tag{1}$$

To consider a wide range of geological material properties that are expected to affect the failure behaviours of rubble piles according to previous studies[50,59–61], experiments were run with four friction angles and a series of cohesion values, as summarised in Supplementary Table 1.

## Modelling of YORP-induced quasi-static spinup

Given that the mechanical performance of granular medium is sensitive to the loading history[62], we need to let the rubble-pile model settle down in a stress-free state and simulate its realistic loading history. Therefore, in our simulations, to precisely capture the spin-driven structural evolution of Bennu, the rubble-pile model is initially rotating at a relaxed spin rate and is spun up slowly consistent with the YORP effect.

We implemented a quasi-static spinup procedure that sets the simulated rubble pile's spin period $T$ to a prescribed value as a function of time $t$ (Supplementary Fig. 1; only shortest-principal-axis rotation is considered in this study). In the beginning of the simulation, the body settles down at a slow spin state where $T = 10$ h. Then, the body's spin period is first linearly decreased to $T = 4$ h with an averaged acceleration rate of about 1500 degrees day$^{-2}$, and is further decreased with an averaged acceleration rate of about 800 degrees day$^{-2}$. The spin period of the simulated body is strictly constrained to the predefined spinup path until structural failure occurs, at which point the body is set to freely evolve under its self-gravity. To enable the detection of some distinct failure behaviours from the simulations, failure is identified at the state when the longest axis of the body changes by 1.5% of the initial value (i.e., about the radius of the smallest particle). The spinup rates were chosen so that the simulations remain computationally expedient and the rubble pile can stay in quasi-equilibrium states during the spinup. Our previous study has confirmed that the evolution of stress-state variables within a stable rubble pile is not sensitive to changes in the spinup rate provided that the loading is quasi-static[51]. Furthermore, the resulting Euler acceleration is two orders of magnitude smaller than the centrifugal acceleration and, thus, is negligible. Therefore, although the adopted spinup rates are much larger than Bennu's actual YORP acceleration rate, the structural response of Bennu to the YORP spin-up effect can be readily modelled.

It has been suggested that the surface topography changes driven by surface mass movement or cratering events could alter the magnitude and direction of the YORP torque[63,64], and the spin evolution of Bennu might therefore deviate from the adopted spinup path. However, our preliminary simulations of Bennu's coupled shape and spin evolution found negligible changes in the YORP torque until global structural failure occurs[65]. If this is the case, Bennu would be spun up in a relatively constant rate; otherwise, due to the randomness of the YORP torque, the spin-rate-doubling timescale would be longer than the current YORP torque predicts. Nonetheless, the structural failure behaviours of Bennu at its spin limit should be independent of the YORP torque magnitude. Therefore, it is justified to use a continuous spinup path for the purpose of this study.

## Dynamical internal slope and cohesion distribution of a rubble pile

In discrete element modelling, all the calculations are performed at the microscopic particle level. However, for geological materials, the concept of frictional and cohesive strength is commonly associated with the macroscopic properties defined within the context of elastic-plastic theory for continuum media[66]. This definition involves variables that cannot be clearly related to the interparticle contact characteristics at the microscopic scale. Based on the homogenisation and averaged-stress-analysis method[10,67], we divide the simulated rubble pile into representative volume elements (RVEs) and use these elements to calculate the stress distribution. The average stress tensor of a local region, e.g., the $j$-th RVE, in a rubble pile can be assessed by averaging the stress tensor for every particle in this RVE,

$$\bar{\boldsymbol{\sigma}}_j^{\text{RVE}} = \frac{1}{V_j^{\text{RVE}}} \sum_{i=1}^{N_j^{\text{RVE}}} \sum_{k=1}^{N_{\text{cont},i}} \mathbf{x}^{i,k} \otimes \mathbf{f}^{i,k}, \tag{2}$$

where $N_j^{\text{RVE}}$ is the total particle number and $V_j^{\text{RVE}}$ is the total volume of RVE $j$. $N_{\text{cont},i}$ is the total contact number of particle $i$. The branch vector $\mathbf{x}^{i,k}$ links the particle centre to the contact point for the $k$-th contact, with the corresponding contact force $\mathbf{f}^{i,k}$. The Cauchy stress tensor for particle $i$ is expressed as the summation of the dyadic product of $\mathbf{x}^{i,k}$ and $\mathbf{f}^{i,k}$. With the derived principal stresses of the average stress tensor, its first invariant $I_1$ and the deviatoric stress $J_2$ for the $j$-th RVE can then be calculated by,

$$\begin{aligned}
I_{1,j}^{\text{RVE}} &= \bar{\sigma}_{j,1}^{\text{RVE}} + \bar{\sigma}_{j,2}^{\text{RVE}} + \bar{\sigma}_{j,3}^{\text{RVE}}, \\
J_{2,j}^{\text{RVE}} &= [(\bar{\sigma}_{j,1}^{\text{RVE}} - \bar{\sigma}_{j,2}^{\text{RVE}})^2 + (\bar{\sigma}_{j,2}^{\text{RVE}} - \bar{\sigma}_{j,3}^{\text{RVE}})^2 + (\bar{\sigma}_{j,3}^{\text{RVE}} - \bar{\sigma}_{j,1}^{\text{RVE}})^2]/6.
\end{aligned} \tag{3}$$

In effect, $I_{1,j}^{\text{RVE}}$ measures the pressure acting on each RVE, and we use this to evaluate the distribution of the dynamical internal pressure (Fig. 3).

When cohesion is not present, the structural stability of each RVE is assessed by the dynamical internal slope $\theta^{\text{inter}}$. Based on the Drucker–Prager failure criterion[66], i.e.,

$$\sqrt{J_2} \le \frac{6C\cos\phi}{\sqrt{3}(3 - \sin\phi)} + \frac{2\sin\phi}{\sqrt{3}(3 - \sin\phi)}I_1, \tag{4}$$

the internal slope of the $j$-th RVE is determined by solving (taking $C = 0$ Pa)

$$\frac{2\sin\theta_j^{\text{inter}}}{\sqrt{3}(3 - \sin\theta_j^{\text{inter}})} = \frac{\sqrt{J_{2,j}^{\text{RVE}}}}{I_{1,j}^{\text{RVE}}}. \tag{5}$$

Accordingly, we can derive the friction angle of a rubble pile by finding the maximum internal slope at its failure spin state, $\phi = \max_{j \in \text{RVEs}} \theta_j^{\text{inter}}|_{\text{failure}}$. An RVE is considered to experience structural failure if its $\theta^{\text{RVE}}$ is larger than the material friction angle $\phi$.

When cohesion is present, the structural stability of each RVE is then assessed by the dynamical internal cohesion $C^{\text{RVE}}$. Based on the same failure criterion, the internal cohesion of the $j$-th RVE is computed by

$$C_j^{\text{RVE}} = \frac{\sqrt{3}(3 - \sin\phi)}{6\cos\phi}\sqrt{J_{2,j}^{\text{RVE}}} - \frac{\tan\phi}{3}I_{1,j}^{\text{RVE}}. \tag{6}$$

Similarly, an RVE is considered to fail if its $C^{\text{RVE}}$ is larger than the material cohesion $C$.

## Dynamical surface slope and cohesion distribution of a rubble pile

For a rubble pile rotating with an angular velocity $\boldsymbol{\Omega}$, the equation of motion for a surface particle with radius $R_{\text{p}}$ located at $\mathbf{r}_{\text{p}}$ in the body-fixed frame can be written as, $\mathbf{a}_{\text{p}} = \mathbf{a}_{\text{p}}^{\text{grav}} + \mathbf{a}_{\text{p}}^{\text{cent}} + \mathbf{a}_{\text{p}}^{\text{Euler}} + \mathbf{a}_{\text{p}}^{\text{Cori}} + \mathbf{F}_{\text{p}}^{\text{cont}}/m_{\text{p}}$, where $\mathbf{a}_{\text{p}}$ and $m_{\text{p}}$ are the acceleration and mass of the particle, respectively. $\mathbf{a}_{\text{p}}^{\text{grav}}$ is the local gravitational acceleration. The centrifugal acceleration $\mathbf{a}_{\text{p}}^{\text{cent}} = -\boldsymbol{\Omega} \times (\boldsymbol{\Omega} \times \mathbf{r}_p)$. The Euler acceleration $\mathbf{a}_{\text{p}}^{\text{Euler}} = -\dot{\boldsymbol{\Omega}} \times \mathbf{r}_{\text{p}}$. We define the sum of these first three terms as the static effective acceleration $\mathbf{a}_{\text{p}}^{\text{eff}}$. When the contact force $\mathbf{F}_{\text{p}}^{\text{cont}}$ is not able to balance the effective acceleration, the particle gains net acceleration and starts to move on the surface. In this case, the Coriolis acceleration, $\mathbf{a}_{\text{p}}^{\text{Cori}} = -2(\boldsymbol{\Omega} \times \mathbf{v}_{\text{p}})$, where $\mathbf{v}_{\text{p}}$ is the particle velocity, could become dominant and even loft the particle above the surface.

When cohesion is not present, we define the dynamical surface slope $\theta^{sur}$ to assess the surface stability. We first compute a tentative slope as the angle between the local surface normal and the effective acceleration, i.e.,

$$\bar{\theta}_p^{sur} = \arccos[(\mathbf{a}_p^{eff} \cdot \hat{\mathbf{n}}_p)/a_p^{eff}], \tag{7}$$

to evaluate the mobility of the surface particle. We construct a polyhedral shape of the given rubble-pile model to calculate $\hat{\mathbf{n}}_p$ based on the alpha-shape algorithm[68]. By varying the alpha radius $\alpha$ from 10 to 50, incremented by 10, we obtained $\alpha = 30$ for best fitting the surface slope distribution of Bennu[69] (Supplementary Fig. 5). With the adopted spinup rate $\dot{\mathbf{\Omega}} = 800–1,500$ degrees day$^{-2}$, $a_p^{Euler}$ is two orders of magnitude smaller than $a_p^{cent}$. Therefore, $\bar{\theta}_p^{sur}$ is almost the same as the commonly used definition of planetary body surface slopes[70].

If $\bar{\theta}_p^{sur} > \phi$, we take the Coriolis acceleration into account and consider a conservative estimate of the particle speed, $v_p = \sqrt{2R_p a_p^{eff} \cos \bar{\theta}_p^{sur} (\tan \bar{\theta}_p^{sur} - \tan \phi)}$, i.e., the speed of an initially static surface particle travelling $R_p$ on the surface. To consider the upper lofting limit with this speed, the direction of $\mathbf{a}_p^{Cori}$ is set to align with the local surface normal $\hat{\mathbf{n}}_p$. Then we use the estimated Coriolis acceleration to check if the particle can be lofted and modify the slope accordingly, i.e.,

$$\theta_p^{sur} = \begin{cases} -1, & a_p^{eff} \cos \theta^{sur} < a_p^{Cori} \text{ (lofted)}; \\ \bar{\theta}_p^{sur}, & \text{otherwise}. \end{cases} \tag{8}$$

When cohesion is present, we define the dynamical surface cohesion $C^{sur}$ for the surface stability assessment. The minimum cohesive force required to prevent a surface particle from moving is given as,

$$m_p a_p^{cohe} = \begin{cases} 0, & \theta_p^{sur} \le \phi; \\ m_p a_p^{eff} \cos \theta_p^{sur}[(\tan \theta_p^{sur})/(\tan \phi) - 1], & \theta_p^{sur} > \phi. \end{cases} \tag{9}$$

According to our cohesive force model and Eq. (1), the dynamical surface cohesion can then be calculated by

$$C_p^{sur} = \frac{m_p a_p^{cohe} N_{cont} \eta_{inter} \tan \phi}{4\pi R_p^2}. \tag{10}$$

A surface particle can be mobilised if $C_p^{sur} > C$. In this case, this particle could be lofted if

$$a_p^{eff} \cos \theta_p^{sur} + 4\pi R_p^2 C/(N_{cont} \eta_{inter} \tan \phi) < a_p^{Cori}, \tag{11}$$

where the particle speed is calculated by $v_p = \sqrt{2R_p[a_p^{eff} \sin \bar{\theta}_p^{sur} - (a_p^{eff} \cos \bar{\theta}_p^{sur} + a_p^{cohe}) \tan \phi]}$.

In the analyses of Figs. 2–4, we adopted the maximum particle radius in our rubble-pile model, i.e., $R_p = 12$ m, for the Coriolis acceleration and the surface cohesion estimation. As $C_p^{sur} \propto R_p$, smaller particles are generally more difficult to loft. These macroscopic particles could be individual boulders or agglomerates of smaller grains that clump together due to physical interlocking and/or chemical attractions. The formation of cohesive clumps in surface mass wasting is commonly observed terrestrially[71], and theoretical analyses have shown that clumps of cm-scale and smaller grains are possible to form on asteroids and may be easier to detach from a surface than their constituent grains[72].

## Theoretical failure conditions

Based on the above analyses, we develop a semi-analytical method to quantify the failure conditions for the four failure types identified in this study. With a given rubble-pile body and the associated alpha-shape model, the theoretical failure condition for each failure type can

be derived from the dynamical surface slope/cohesion or the stress state distribution of the body. The failure mode of this body with given material properties can be then diagnosed from a diagram constructed from these theoretical failure conditions (see Fig. 4 and Supplementary Fig. 6 for example), without running any N-body simulations. The procedures to calculate the failure conditions are as follows.

For the Type I surface landslides, we use the dynamical surface slope/cohesion (i.e., Eqs. (7) and (10)) to evaluate the failure condition for the given rubble-pile body. Based on the global surface slope/cohesion distribution (see Supplementary Fig. 5 for an example of the slope distribution), the $2\sigma$ value above the median is adopted as a measure of the maximum surface slope/cohesion. The Type I failure condition is then defined as the spin state where this value exceeds the friction angle or cohesion of the rubble pile.

For the Type III mass shedding, we use the dynamical surface slope/cohesion with consideration of the Coriolis acceleration $a_p^{Cori}$ to evaluate the failure condition, which is defined as the spin state where the inequality Eq. (11) holds.

For the Type II internal deformation and Type IV tensile disruption, we use the semi-analytical stress model introduced by Hirabayashi et al.[21,73] to evaluate the structural stress state distribution and the failure conditions. The Poisson's ratio is set to 0.25. Bennu is approximated as its best-fit ellipsoid[16] with semi-major axes of 252.78 m × 246.20 m × 228.69 m. The Type II failure condition is defined at the spin state where the centre region of the rubble pile violates the Drucker-Prager failure criterion (Eq. (4)), and the Type IV failure condition is defined at the spin state where the stress states of the surface region is in tension (i.e., the dynamical internal pressure $I_1 < 0$).

The failure modes for the Bennu-shaped rubble piles derived from our numerical simulations (Supplementary Table 1) show the same failure-mode-transition trend with the material properties as those indicated by the theoretical failure-mode diagrams (Fig. 4 and Supplementary Fig. 6), and the results are quantitatively matched in the case of low cohesion. For relatively high cohesion, the condition for the Type III mass shedding may be underestimated as we adopted a conservative estimate of the particle speed for evaluating the Coriolis acceleration, and surface particles could be ejected when they lost cohesive contacts with the subsurface layer due to the high centrifugal acceleration. Therefore, in the case of $\phi = 35°$ and $\phi = 40°$, mass shedding is ubiquitously detected in the numerical simulations.

## Avalanche numerical experiments

To make direct comparisons with Bennu's surface slopes, we carried out numerical simulations of avalanches under Bennu's microgravity condition (i.e., $5 \times 10^{-6}g$, where $1g$ is the Earth gravity) to derive the angles of repose of an equivalent granular bed. The term "equivalent" means that the particle size distribution of the granular bed follows the same power law as the Bennu-shaped rubble-pile model. The granular bed has a dimension of 3 m (length) × 2.5 m (width) × 2 m (depth), and the particle size ranges from 2 cm to 6 cm, which represents a local surface area on Bennu consisting of cm-sized grains. The total particle number is about 60,000. The granular bed is settled down under $5 \times 10^{-6}g$. During the avalanche experiment, the inclination angle of the granular bed is gradually increased from 0° up to 60° by an interval of 1° every 10 min. The repose angle is the maximum inclination angle before apparent landslides are detected. With the low-friction interparticle parameter set of $(\mu_S, \beta) = (0.2, 0.3)$, we found that, when cohesion is not present, the angle of repose is close to the macroscopic friction angle $\phi$, i.e., 29°. With a tiny amount of interparticle cohesion of $c = 1$ Pa, whose corresponding macroscopic cohesion $C \approx 0.013$ Pa, the angle of repose increases to 35°. With $c = 3$ Pa, for which $C \approx 0.038$ Pa, no avalanche and apparent surface movement are detected at the final inclination angle of 60°.

## Data availability

The OSIRIS-REx Laser Altimeter (OLA) Bennu shape model (v20) is available from the Small Body Mapping Tool (SBMT) at https://sbmt.jhuapl.edu/Object-Template.php?obj=77. The datasets generated during and/or analysed during the current study are available. Source data are provided with this paper. All necessary files to reproduce the outcome of the 1st, 16th and 24th cases given in Supplementary Table 1 and shown in Fig. 2, are provided with the Supplementary Software file, as examples. The input data for other cases presented in this study are available from the corresponding author on reasonable request. Source data are provided with this paper.

## Code availability

The simulations are performed by the code PKDGRAV in its custom version that includes the Soft-Sphere Discrete Element Method (see Methods). Two compiled versions of PKDGRAV (one with the spinup module and one without) are provided as Supplementary Software 1. Visualisation support is provided by the open-source POV-Ray ray-tracing package (https://www.povray.org; Fig. 1a and all the supplementary movies).

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

## Acknowledgements

Y.Z. and P.M. acknowledge funding from the French space agency CNES, the European Union's Horizon 2020 research and innovation program under grant agreement No. 870377 (project NEO-MAPP), and the Doeblin Federation (project Impact-Granular Simulations). O.S.B. was supported by the NASA Solar System Working Program under grant number NNX16AQ13G. K.J.W. was supported by the NASA Solar System Exploration Research Virtual Institute node Project ESPRESSO, under the cooperative agreement number 80ARC0M0008. Simulations were performed on Mésocentre SIGAMM hosted at the Observatoire de la Côte d'Azur and on the Deepthought2 HPC cluster administered by the Division of Informational Technology at the University of Maryland. We are grateful to the entire OSIRIS-REx Team of engineers, operators, scientists and administrators for making the encounter with Bennu possible.

## Author contributions

Y.Z. implemented the spinup module in the PKDGRAV code, conducted the soft-sphere *N*-body simulations, constructed the analytical formalism to assess rubble-pile failure modes, and led the conceptualisation of the study, the interpretation of the results, and manuscript writing. P.M. contributed to the conceptualisation of the study, simulation parameter space definition, the interpretation of the results, and manuscript writing. O.S.B. contributed to the conceptualisation of the study and the comparisons of the numerical results with Bennu's surface features. J.H.R., M.G.D., R.L.B., K.J.W., and C.M.H. contributed to the comparisons of the numerical results with Bennu's surface features. D.C.R. contributed to the implementation of the spinup module in the PKDGRAV code and the interpretation of the results. D.S.L. leads the OSIRIS-REx mission and contributed to the comparisons of the numerical results with Bennu's surface features. All authors contributed to the preparation of the manuscript.

## Competing interests

The authors declare no competing interests.
