## [Peer Review File · Nature Communications]

Inferring interiors and structural history of top- shaped asteroids from external properties of asteroid (101955) BennuEditorial Note: Parts of this peer review file have been redacted as indicated to avoid any copy right infringement.

REVIEWER COMMENTS

Reviewer #1 (Remarks to the Author):

This paper presents results from numerical simulations concerning a particular type of asteroid shape, commonly referred to as 'top-shaped'. It is widely accepted that these distinctive shapes are caused by thermal radiative torques, i.e. the YORP effect (although, some in the field attribute these shapes to collisional processes only, but this is highly unlikely).

These top-shaped asteroids have now been visited by three spacecraft (first by Rosetta with asteroids Steins, and later by OSIRIS-Rex and Hayabusa 2). Given this, and the increasing number of such objects being discovered by ground-based planetary radar observations, it is clear that such objects are ubiquitous in the inner solar system. As such, it is particularly important to understand how YORP shaped them, and what we can potentially say about their interiors.

The study aims to do that by simulating the structural behaviour of objects with this shape under the influence of YORP, for a range material properties, specifically 'friction angles' and cohesion. What makes the study particularly valuable, is that the authors have created asteroid simulants comprised of 'soft-spheres' with relatively high resolutions (~41,000 particles), whereby the material properties can be varied, as noted above. This is a difficult and time-consuming task.

I have some brief and general comments about the paper. Overall, I find the work very interesting indeed, including the conclusions drawn. The techniques used, and their implementation, seem sound. However, I have some difficulty relating the conclusions listed in the abstract with the main text, and determining if they are fully justified. The conclusion concerning carbonaceous asteroids (lines 33-35), is particularly problematic (in other words, can the authors really be this specific?). The presentation style of the study leads to it being somewhat difficult to disentangle the various scenarios explored and related conclusions, from each other (and the description of how the parameter space was explored could be simplified). In my opinion, the paper is not as accessible as it could be, especially for a paper in one of the Nature journals. Before the study can be published, perhaps the authors should address this.

Other minor, and more specific, comments include:

- It would be useful to see some text added to the start of the main text, covering similar studies, and how the current study compares with the state of the art, and precisely how the results presented here advance the situation.

- Please also mention the flyby of main belt asteroid Steins, by Rosetta (also see Keller et al, Science 2010). That was the first object to be visited by spacecraft that was clearly top-shaped and evolved from YORP.

- In the abstract, it is noted that no 'direct' measurements concerning asteroid interiors has been made. Strictly speaking, this is true, but there are strong 'indirect' indications of internal density inhomogeneity for asteroid Itokawa (see Lowry et al. A&A 2014), which is worth mentioning. The density differences noted between the 'head' and 'body' of Itokawa are consistent with other studies on the formation (and subsequent collapse) of binary systems, and recent work on re-accumulation of fragments from catastrophic collisions, which can produce Itokawa-like bodies with precisely the same density ratios noted in Lowry et al.

- The figures could be greatly simplified, especially Figure 3. There is simply too much information there, that isn't well described, and not well linked with the main text.

=====

Reviewer #2 (Remarks to the Author):

Comments on the manuscript "External properties as a guide to interiors and structural histories of top-shaped asteroids" by Y. Zhang et al.

As usual with Mrs. Zhang papers, it is a pleasure to read them because they are very well written. The methods and results in the manuscript are presented clearly and consistently.

What are the noteworthy results?

Based on numerical simulations of granular media sustained by self-gravity, the authors obtained relevant constraints to the internal structure and material strength of asteroids, like Bennu.

Will the work be of significance to the field and related fields?

The manuscript is a significant contribution on a elusive topic: the interior of asteroids. Many a space missions (like OSIRIS-REx, Hayabusa 2, Lucy, Psyche, Hera) have among the main scientific objectives to provide information about this issue. Therefore, the results presented in this manuscript are useful to interpret the data from these missions.

How does it compare to the established literature?

The manuscript is a new contribution in the series of papers by Y. Zhang (2018, 2021) about rotational effects and stability of rubble-pile asteroids. In this paper, as well as in the others mentioned before, the authors presented a comprehensive review of the literature on the subject.

Does the work support the conclusions and claims?

Yes, the simulations and theoretical work give strong support to the presented results.

Are there any flaws in the data analysis, interpretation and conclusions?

There are only a few clarifications points listed below.

Is the methodology sound?

Yes, the authors have successfully used the numerical simulations using soft-sphere methods for several years.

Does the work meet the expected standards in your field?

Yes, it is a relevant contribution.

Is there enough detail provided in the methods for the work to be reproduced?

The numerical simulations were done with pkdgrav code, which is not publicly available. Nevertheless, is a package that has been used by many colleagues. In addition, those simulations can be reproduced with other DEM packages, some of them are open source.

I have just very minor comments:

page 3, line 82:

Is the sentence: "the body can enter the right-hand sides of the Type I curves in Fig. 3..." correct?

Or should it be "the left-hand sides"?

Otherwise, I do not understand the process.

page 4, line 103

Although the layered model is briefly explained in the caption of the "Supplementary Table 1", I suggest to describe the model more explicitly in the Methods.

page 13, line -6 from bottom

"Therefore, rubble piles can across the Type I failure curves ..."

There is a missing verb, because across is a preposition or an adverb.

page 17, between 343 and 344

Could the authors show where the value of ϕ is derived?

page 26, Supplementary Table 1

Is C in this table the macroscopic cohesion, or the interparticle tensile strength?

Because in the Table header is presented as C , but in the caption it is referred as c .

Congrats for this new contribution.

Gonzalo Tancredi

Reviewer #3 (Remarks to the Author):

This is an interesting paper, the authors have simulated an idealised version of asteroid Bennu, the target of the OSIRIS-REx mission and, based on the surface features, or lack thereof, they have inferred a possible interior structure. The paper is well written and easy to follow and the conclusions are compelling. The one complaint I have is that the authors have disregarded the previous work carried out by many other scientists. Many of the things that the authors “find” had already been found years ago; however, there isn’t a single reference to previous work that has specifically dealt with certain issues about rotational disruption of small asteroids. The authors should include clear reference to previous work and differentiate between what they have truly found and what has been simply corroborated by them.

This being the case, I cannot recommend this paper for publication in Nature Communications. Though the paper seems scientifically sound, that is not the only criterion that has to be satisfied. This paper has analysed a simulated version of Bennu and has found that much of what has been found for idealised self-gravitating bodies is corroborated for this specific case. Much of it is not new but an application of previously acquired knowledge. “In general, to be acceptable, a paper should represent an advance in understanding likely to influence thinking in the field, with strong evidence for their conclusions.” and this paper does not accomplish any but the last of all these points. Regardless of this, I would like to ask the authors to resubmit the paper to another, more appropriate journal specific to the field. If you choose to do so, please take into account the following comments.

Major Comments:

- Title: You are mainly talking about Bennu, this paper is not really about two asteroids, but one. Please change your title to reflect this.

- In 68: It seems to me you are talking about Reynolds dilation. This phenomenon is common to granular materials, but it depends on the initial condition of the medium. This was also observed by Sanchez and Scheeres (2012). This is, if it is all well compacted, the volume will have to increase to allow failure; however, if the initial state of the medium is very loose, it will continually compact under shear. If you only see dilation, it seems to me you always start from a well compacted structure which might not be realistic for an asteroid that is formed under micro-gravity conditions. Did you do some experiments with a more porous structure? Why not?

- In 69-70: This was first observed by Hirabayashi (2015) "Failure modes and conditions of a cohesive, spherical body due to YORP spin-up."

- In 74-78: There are no details about this, so I will assume that your cohesive aggregates are completely homogeneous. This observation, that the core of a self-gravitating aggregate is going to fail before the surface and that it will need to be stronger to be stable at elevated spin rates was first observed and reported by Hirabayashi, et al, Internal structure of asteroids having surface shedding due to rotational instability (2015). Then this was further studied by Sanchez and Scheeres (2018), Rotational evolution of self-gravitating aggregates with cores of variable strength.

These studies certainly worked with spherical and prolate shapes and you have chosen the shape of a real asteroid, but if the observation is exactly the same, you should acknowledge that and point to the fact that it seems to be a more general outcome than previously thought.

- In 79: You have not discovered these patterns in the general sense, you have observed them for a particular shape as they were already discovered by the previously cited authors.

- In 100-104: Hirabayashi, Sanchez and Scheeres have worked on this specific topic for years, I find it very strange that most of their papers are not cited. The inclusion of strong and weak cores was the subject of some of the papers I have cited above. You would do well in getting acquainted with their results.

- In 123-124: Or it could be that, though maybe not as likely, as proposed by Tardivel, et al (2018), "Equatorial cavities on asteroids, an evidence of fission events", a cohesive chunk of material simply was detached from the equatorial region. Have you completely discarded this possibility? Why?

- In 141-142: Again, this model was first explored, proposed and implemented by Sanchez and Scheeres (2014), The strength of regolith and rubble pile asteroids. However, it has not been cited.

- In 152-154: This could be true; however, it does not explain its symmetry. Can you explain the symmetric, "squarish equatorial shape?"

- In 158-159: Again, the core could indeed be more porous or weaker than the surface, but this cannot be readily linked to the internal shear deformation. As explained above, this decrease in filling fraction happens only when the initial aggregate is well packed. This is the starting point you have chosen for your simulations, but you have not justified why this would be a realistic starting

point for Bennu. Furthermore, if this is an unlikely starting point, I think most of what you have observed in your simulations could be inapplicable to Bennu.

- In 161-162: What do you mean by your approach? Are you talking about using SSDEM simulations to form self-gravitating aggregates with different material and bulk characteristics to then spin them up and associate their findings to observed features in small bodies? Scientists have been doing numerical simulations using soft-sphere DEM codes since 2008 when Sanchez and Scheeres introduced them to Planetary Sciences. Richardson, Michel and Schwartz did this using a HSDEM code. Hirabayashi, Rozitis and Holsapple have been doing theoretical and numerical work for the last 2 decades. Please ensure to cite all relevant literature and consider summarizing previous contributions to the field.

- In 165-166: This was expressed in Sanchez and Scheeres (2014) and expressly used in their 2018 paper about cores of variable strength.

- In 169-170: This statement makes no sense unless you cite Sanchez and Scheeres (2014) about the strength of rubble pile asteroids.

- fig 4: Compare to the images in Hirabayashi, et al, Internal structure of asteroids having surface shedding due to rotational instability (2015), or Sanchez and Scheeres (2018), Rotational evolution of self-gravitating aggregates with cores of variable strength. In them, they explain how the strength of the core is directly linked to the failure mode of the internal structure. Please, cite those earlier studies, discuss potential overlaps and stress how your study advances those.

- In 321: The Soft-Sphere method was first implemented for the simulation of small bodies by Sanchez and Scheeres, the numerical method itself was first implemented for granular materials by Cundall and Strack. The authors seem to imply here that this method is their invention, which it is most certainly not. Please refer to those relevant literature.

- In 354: This figure does not really contain anything to help me see that the process is quasi-static. Please, add another figure.

- In 359-361: Structural failure, though it is linked to body deformation, is directly detected by a yield criterion. This has been extensively studied in mechanical and civil engineering. Its application to asteroids has been carried out mainly by Sharma, Holsapple and Hirabayashi for the last two

decades to mention a few. Please, use that; you have the tools to calculate the stress tensor which is what you need as input for the yield criterion calculation.

- In 370-373: I think it is Ok to cite preliminary work and results, but I don't think it is appropriate to say that these results fully justify to do something without any caveats. If these results are preliminary, it means that a complete study is not yet finished and your conclusions could change. Please, either remove or change this statement to reflect this.

- In 396-398: This analysis is similar to that carried out by Patel and Hartzell (2021), "A Model to Predict the Size of Regolith Clumps on Planetary Bodies." Please, see if a citation is needed here or if you need to modify this section to use their work. If not, please explain why this is better, or more applicable, than what they had done.

- In 442: This is the kinetic angle of repose.

- In 443-445: Yes, theoretically these two should be equal; this is a known result.

Minor Comments:

- In 50: Please, provide the particle size range.

- In 121: "... Bennu's surface is due to..." -> "... Bennu's surface could be due to..."

- In 335: It seems to me that you are using a linear spring-dashpot model here. If you are, please be explicit about it as non-linear models could also be used.

- In 396-397: "... a initially..." -> "... an initially..."

In this report, I have tried to highlight obvious omissions to previous work to make it obvious that the paper does not fulfill the criterion for publication of Nature Communications. This is not something that can be solved just by adding some citations as, even if added, this will simply highlight the fact that the paper is an application, albeit an interesting one, of previously known

results. As such, it is my belief that the paper does not “represent an advance in understanding likely to influence thinking in the field.” To show this in a simple detail, the following conclusion can be found in Sanchez and Scheeres (2018), “Rotational evolution of self-gravitating aggregates with cores of variable strength”: “This implies that the specific shapes, spin rates and surface features that have been observed in asteroids are an expression of their hidden interiors and, as such, should provide insight about the formation and evolution processes of these bodies,” which is the main idea of the title and the paper. Yet, this paper is never cited.

Apart from this, the authors do not justify their choice of initial packing and this carries many implications for the subsequent evolution. How this affects the results is not a trivial matter, but the authors ignore this and simply apply all their results to Bennu with no caveats at all. In fact, a change in the initial packing of the body could lead to a completely different evolution.

Response Letter to Reviewers

We thank the reviewers for their insightful comments and corrections.

Below, we address these comments and concerns (presented **in bold**) point by point and highlight the main corrections in the manuscript **in red colour** if they occur. All the changes are also highlighted **in red colour** in the revised manuscript. As we use the custom code *pkdgrav* in the current study, the pre-compiled executable of *pkdgrav* and three example cases (see “readme.md” in the uploaded “code_pkdgrav.zip” file) are provided for reviewer assessment.

Reviewer #1:

General comment: This paper presents results from numerical simulations concerning a particular type of asteroid shape, commonly referred to as ‘top-shaped’. It is widely accepted that these distinctive shapes are caused by thermal radiative torques, i.e. the YORP effect (although, some in the field attribute these shapes to collisional processes only, but this is highly unlikely).

These top-shaped asteroids have now been visited by three spacecraft (first by Rosetta with asteroids Steins, and later by OSIRIS-Rex and Hayabusa 2). Given this, and the increasing number of such objects being discovered by ground-based planetary radar observations, it is clear that such objects are ubiquitous in the inner solar system. As such, it is particularly important to understand how YORP shaped them, and what we can potentially say about their interiors.

The study aims to do that by simulating the structural behaviour of objects with this shape under the influence of YORP, for a range material properties, specifically ‘friction angles’ and cohesion. What makes the study particularly valuable, is that the authors have created asteroid simulants comprised of ‘soft-spheres’ with relatively high resolutions (~41,000 particles), whereby the material properties can be varied, as noted above. This is a difficult and time-consuming task.

I have some brief and general comments about the paper. Overall, I find the work very interesting indeed, including the conclusions drawn. The techniques used, and their implementation, seem sound. However, I have some difficulty relating the conclusions listed in the abstract with the main text, and determining if they are fully justified. The conclusion concerning carbonaceous asteroids (lines 33-35), is particularly problematic (in other words, can the authors really be this specific?). The presentation style of the study leads to it being somewhat difficult to disentangle the various scenarios explored and related conclusions, from each other (and the description of how the parameter space was explored could be simplified). In my opinion, the paper is not as accessible as it could be, especially for a paper in one of the Nature journals. Before the study can be published, perhaps the authors should address this.

Response: We thank the reviewer for acknowledging the scientific importance and soundness of our study. To clarify the conclusion and improve the accessibility of the manuscript, we have made the following revisions.

The original conclusion, i.e., “*The reconfiguration pathways inferred here imply that satellite formation via mass shedding for carbonaceous asteroids is inhibited by their interior and structural properties.*”, listed at the end of the abstract, was meant to state that if carbonaceous asteroids have similar material properties as those of Bennu, they cannot have mass shedding failure behaviours and the subsequent satellite formation via this path is prohibited. To make the description more clear and justified, we modified the last sentence of the abstract to, “**Furthermore, we reveal the underlying mechanisms that lead to different failure behaviours and identify the reconfiguration pathways of top-shaped asteroids as functions of their properties that either facilitate or prevent the formation of moons.**”, in the revised manuscript.

To improve the accessibility, we have: 1) added an **Introduction Section** to introduce the background and motivation, the state-of-the-art, the present challenges, the study conception, the modelling approach, and the explored parameter space; 2) restructured the results into three main sections in the **Results Section**, i.e., “**Typical failure behaviours of top-shaped rubble piles**”, “**A unified phase diagram for failure mode diagnosis**”, and “**Structural properties and evolution of Bennu**”, to disentangle the failure mode analyses from the analyses about possible structural properties and evolution scenarios of Bennu; 3) summarised the main conclusions at the beginning of the **Discussion Section** and moved the discussions about our study’s implications to this section.

For the sake of brevity, these revisions are not produced in this letter but are all highlighted in red colour in the revised manuscript.

Minor comment 1: It would be useful to see some text added to the start of the main text, covering similar studies, and how the current study compares with the state of the art, and precisely how the results presented here advance the situation.

Response: We thank the reviewer for the comment. We have restructured the manuscript and added an **Introduction Section** to state the motivation of the current study and the current state of the art. In particular, to cover similar studies and present the challenge and how the results of the current study can advance the situation, we added a new paragraph starting from Line 36, i.e., “**To date, little is known about the internal structure and material properties of asteroids except for some indirect indications regarding the internal density distribution (Lowry et al. 2014; Scheeres et al. 2020). Previous studies showed that the reshaping process of a spherical rubble-pile body under YORP spinup would be altered by the amount and distribution of material properties within this body (Hirabayashi et al. 2015; Sánchez & Scheeres 2018; Zhang et al. 2018; Ferrari & Tanga 2022) but lacked connections with actual asteroid surface data. The next challenge is to use detailed geophysical features returned by spacecraft equipped with state-of-the-art instruments to constrain or even identify the interior properties as well as the associated structural history of visited asteroids. This provides an opportunity to infer evolutionary scenarios from these asteroids’ parent-body disruption to their current states along with important insights for future asteroid space exploration. In this study, we tackle this challenge.**”, in the revised manuscript.

Following this new paragraph, we then elaborated the complex geophysical features of Bennu returned by the OSIRIS-REx mission, and added the following discussions (starting from Line 66) to highlight the difficulty of linking theoretical/numerical analyses with actual asteroid surface data and present our approach, “**Interpretation of this complex set of information requires a comprehensive understanding of asteroid geophysical processes and their proper modelling. Here, to connect all the clues together and infer Bennu’s reshaping history in a self-consistent scenario, we performed numerical simulations using the Soft-Sphere Discrete Element Method (SSDEM; Sánchez & Scheeres 2011, Schwartz et al. 2012) to test Bennu’s structural response to YORP-induced spinup.**”, in the revised manuscript.

At the end of the **Introduction Section**, we added a sentence to highlight the main results and show how the goal of tackling the mentioned challenge is achieved, i.e., “**By exploiting the numerical results and making comparisons with Bennu’s geophysical features, we quantify the interior properties and derive a unified evolutionary scenario for Bennu, which allows us to draw general implications for the structural evolution of top-shaped rubble piles.**”, in the revised manuscript.

Minor comment 2: Please also mention the flyby of main belt asteroid Steins, by Rosetta (also see Keller et al, Science 2010). That was the first object to be visited by spacecraft that was clearly top-shaped and evolved from YORP.

Response: The Rosetta mission’s target asteroid Steins is mentioned along with Ryugu and Bennu as the three examples of top-shaped asteroids visited by spacecraft in Line 46–48, i.e., “Three top-shaped asteroids have been visited by spacecraft, i.e., (2867) Steins by the ESA’s Rosetta mission (Keller et al. 2010), (162173) Ryugu by the JAXA’s Hayabusa2 mission (Watanabe et al. 2019), and (101955) Bennu by NASA’s Origins, Spectral Interpretation, Resource Identification, and Security-Regolith Explorer (OSIRIS-REx) mission (Lauretta et al. 2017).”, in the revised manuscript.

Minor comment 3: In the abstract, it is noted that no ‘direct’ measurements concerning asteroid interiors has been made. Strictly speaking, this is true, but there are strong ‘indirect’ indications of internal density inhomogeneity for asteroid Itokawa (see Lowry et al. A&A 2014), which is worth mentioning. The density differences noted between the ‘head’ and ‘body’ of Itokawa are consistent with other studies on the formation (and subsequent collapse) of binary systems, and recent work on re-accumulation of fragments from catastrophic collisions, which can produce Itokawa-like bodies with precisely the same density ratios noted in Lowry et al.

Response: We added a sentence in the Introduction Section to discuss the indirect indications asteroid interiors, i.e., “To date, little is known about the internal structure and material properties of asteroids except for some indirect indications regarding the internal density distribution (Lowry et al. 2014; Scheeres et al. 2020).”, in Line 36–37 in the revised manuscript.

Minor comment 4: The figures could be greatly simplified, especially Figure 3. There is simply too much information there, that isn’t well described, and not well linked with the main text.

Response: We acknowledge that Figure 3 is complex and difficult to read. We would like to keep this figure as it is to not lose information. Instead, we added more description to guide the reader to walk through the spinup-reshaping process presented by this diagram in the main text of the revised manuscript.

In the paragraph starting from Line 124, we added the following description in the beginning, “The limiting spin period that can activate each failure mode in a rubble pile with given material properties is plotted in Fig. 3. Since the failure condition is evaluated based on the stress state at different failure locations for different failure modes and the stress state in a rubble pile varies significantly within the body (see Fig. 2 for examples), the limiting spin periods of the body for different failure modes are substantially different from each other. During a spinup process (from left to right in Fig. 3a, b), the body will exhibit the corresponding failure behaviour when its spin period decreases to the limiting spin period of a certain failure mode, i.e., by reaching the corresponding failure curve.”, to explain what the curves in this figure indicate.

In the paragraph starting from Line 137, we added the following description in the beginning, “The horizontal dashed lines with arrowheads in Fig. 3 illustrate the spinup process and the failure mode of a rubble pile with given material properties.”, and then elaborated the spinup path and evolutionary scenario of each case to guide the readers.

Once again, thank you very much for your insightful comments and suggestions.

Reviewer #2:

We thank the reviewer for the extremely positive general comments (which are not reproduced here for the sake of brevity). Please see below for our responses to the specific comments.

Minor comment 1: page 3, line 82: Is the sentence: "the body can enter the right-hand sides of the Type I curves in Fig. 3 ..." correct? Or should it be "the left-hand sides"? Otherwise, I do not understand the process.

Response: This should be the same as the spin period decreasing direction, i.e., the right-hand sides. This is due to the fact that Type I surface landslides only occur at some high-slope regions, and so, when a rubble pile is spun up by the YORP effect, it can keep its structure stable at faster spin by reducing the slope in these local surface regions via landslides. To clarify, we modified the original description to, "As landslides only occur locally at some high-slope regions and the slopes at these regions can be reduced concurrently with the landslides (Barnouin et al. 2021), the body can enter the right-hand sides of the Type I curves in Fig. 3 when it is spun up by the YORP effect.", in Line 131–133 in the revised manuscript.

Minor comment 2: page 4, line 103: Although the layered model is briefly explained in the caption of the "Supplementary Table 1", I suggest to describe the model more explicitly in the Methods.

Response: We added the following description to introduce the layered model, "In a homogeneous structure, a constant c is used throughout the rubble-pile model; in a layered structure, c is set to 0 Pa in the 25-meter-depth surface layer and nonzero values in the interior; in a heterogeneous structure, only the specified cohesive regions have nonzero c and particle interactions in other locations are cohesionless.", in the third paragraph of Methods Section "Soft-sphere discrete element modelling" (in Line 334–338) in the revised manuscript.

Minor comment 3: page 13, line -6 from bottom: "Therefore, rubble piles can across the Type I failure curves ..." There is a missing verb, because across is a preposition or an adverb.

Response: Thank you for pointing this out. The word "across" is replaced by "cross" in the revised manuscript.

Minor comment 4: page 17, between 343 and 344: Could the authors show where the value of ϕ is derived?

Response: We added the following description in Line 342–349, "In the case where a rubble pile fails through internal deformation, we can derive its friction angle by finding the maximum internal slope according to Eq. (5) (see below) at its failure spin state, $\phi = \max_{\phi \in \text{RVEs}} \phi^{\text{inter}}_{\text{failure}}$. In the case where it fails through surface mass shedding, ϕ is taken to be the angle of repose derived from our avalanche numerical experiments (see below). Although the angle of friction and the angle of repose may differ due to the different confining pressure (Al-Hashemi & Al-Amoudi 2018), by taking the case of $(\phi_{\text{int}}, \phi) = (0.2, 0.3)$ as an example, our avalanche numerical experiments show that the internal friction angle is close to the angle of repose under the microgravity environment of Bennu.", to explain how we measure the friction angle ϕ in the revised manuscript.

Minor comment 5: page 26, Supplementary Table 1: Is C in this table the macroscopic cohesion, or c the interparticle tensile strength? Because in the Table header is presented as C , but in the caption it is referred as c .

Response: Yes, that is correct. C (the seventh column) is the macroscopic cohesion, and c (the third column) is the interparticle tensile strength. To clarify, we modified the caption of Supplementary Table 1 to, "Material

properties (μ , C), critical spin period (T_{crit}), and failure mode under spinup loading of Bennu-like rubble-pile model with different interparticle parameter sets (μ , C , and c), in the revised manuscript.

Once again, thank you very much for your insightful comments and suggestions.

Reviewer #3:

General comment: This is an interesting paper, the authors have simulated an idealised version of asteroid Bennu, the target of the OSIRIS-REx mission and, based on the surface features, or lack thereof, they have inferred a possible interior structure. The paper is well written and easy to follow and the conclusions are compelling. The one complaint I have is that the authors have disregarded the previous work carried out by many other scientists. Many of the things that the authors “find” had already been found years ago; however, there isn’t a single reference to previous work that has specifically dealt with certain issues about rotational disruption of small asteroids. The authors should include clear reference to previous work and differentiate between what they have truly found and what has been simply corroborated by them. This being the case, I cannot recommend this paper for publication in Nature Communications. Though the paper seems scientifically sound, that is not the only criterion that has to be satisfied. This paper has analysed a simulated version of Bennu and has found that much of what has been found for idealised self-gravitating bodies is corroborated for this specific case. Much of it is not new but an application of previously acquired knowledge. “In general, to be acceptable, a paper should represent an advance in understanding likely to influence thinking in the field, with strong evidence for their conclusions.” and this paper does not accomplish any but the last of all these points. Regardless of this, I would like to ask the authors to resubmit the paper to another, more appropriate journal specific to the field. If you choose to do so, please take into account the following comments.

Response: We thank the reviewer for acknowledging the scientific relevance of our study. However, we totally disagree with the reviewer’s statements, “*Much of it is not new but an application of previously acquired knowledge*”. We believe that this study represents an important advance in understanding asteroid interiors and their structural evolutions. This will be demonstrated by the following three main points in the current response and our answers to the specific comments below.

First, our study advances the modelling and data analysis techniques into new realms that can incorporate and be directly compared with the geophysical features returned by spacecraft equipped with state-of-the-art instruments. Our SSDEM simulations use a high-resolution rubble-pile model (with $N \sim 41,000$ particles) with relatively large particle size differences ($R_{\max}/R_{\min} = 3$) to mimic the observed boulder distribution on Bennu and capture the shape of Bennu to a relatively high resolution (please see Methods Section “Bennu rubble-pile model” for details). The surface slope distribution of our rubble-pile model resembles that of Bennu (see Supplementary Figure 4), which allows us to carry out the surface mass movement analyses. As revealed by the results shown in Figure 1, for appropriate surface material properties, the surface landslide event observed in our simulations can reproduce the mass movement distribution observed on Bennu (which in turn allows us to constrain the surface material properties of Bennu). This high-resolution model also allows us to design the analytical approach (see Methods Section “Dynamical internal slope and cohesion distribution of a rubble pile” and “Dynamical surface slope and cohesion distribution of a rubble pile” for details) to calculate the internal stress- state and surface slope/cohesion distribution (see Figure 2 for examples), which helps us to reveal the underlying mechanisms that lead to different failure behaviours. None of these analyses can be achieved without a high- resolution rubble-pile model, and, thanks to our high-efficiency parallel N -body tree code `pkdgrav`, we can conduct these expensive numerical simulations within acceptable computational costs. All of the previous SSDEM studies mentioned by the reviewer used rubble-pile models that consist of $N \sim 3,000$ particles with size differences $R_{\max}/R_{\min} < 1.5$. These low-resolution models can neither capture the high-accuracy shape model of a space mission target like Bennu nor reproduce the target’s geophysical features. Furthermore, the analytical method and results on the surface mass movement (Figure 1) and stress/slope distribution (Figure 2) have never been reported in any of these previous studies.

Second, our study derives new insights into the surface and internal properties of top-shaped asteroids and their geophysical evolution. Our SSDEM simulations reveal the high sensitivity of a top-shaped rubble-pile asteroid's structural failure behaviours to its structural properties in a quantitative way (see Supplementary Table 1), and our original stress & slope analytical approach explains the mechanisms that induce these behaviours. These two perspectives inspired us to develop a semi-analytical method to quantify the relation between the failure conditions and the structural material properties (see Figure 3). This semi-analytical method can be used without running any SSDEM simulations, and, therefore, is beneficial for a large number of researchers who want to assess the structural history of an asteroid with known shape. It should be noted that there are fundamental differences between our findings and the previous findings on the failure patterns of rubble-pile asteroids mentioned by the reviewer (please see our response to Major comment 4 below for details).

Third, our study, for the first time, develops a unified evolutionary scenario of Bennu that consistently and simultaneously accounts for all its known geophysical characteristics. By numerically modelling Bennu's internal structure, we show that the surface features of Bennu can only be explained by a low-cohesion and relatively low-friction interior with several local high-cohesion regions or solid aggregates (see Figure 4). Moreover, by studying how its internal structure evolves, we find that non-uniform internal deformation induced by this soft heterogeneous interior may have led Bennu to its current squarish-top shape and density distribution and concurrently preserves the old topographic features on its equator. Apparently, none of the previous studies mentioned by the reviewer have considered such a heterogeneous internal structure. In fact, all the previous studies mentioned by the reviewer were published before the OSIRIS-REx spacecraft arrived at Bennu. It is clear that no research so far has reported a model that can systematically account for Bennu's known geophysical characteristics. Furthermore, no study so far has reported an approach that can synthesise geological observations with interior modelling as presented in this paper. This approach can be applied to study other space mission target asteroids, and our results have broad implications for rubble-pile asteroids in general.

To clarify how our study differs from the previous works and how the results presented in the current study can advance the understanding of asteroid interiors and their structural evolutions, an **Introduction Section** is added at the beginning in the revised manuscript. In particular, we added a new paragraph starting from Line 36, i.e., “**To date, little is known about the internal structure and material properties of asteroids except for some indirect indications regarding the internal density distribution (Lowry et al. 2014; Scheeres et al. 2020). Previous studies showed that the reshaping process of a spherical rubble-pile body under YORP spinup would be altered by the amount and distribution of material properties within this body (Hirabayashi et al. 2015; Sánchez & Scheeres 2018; Zhang et al. 2018; Ferrari & Tanga 2022) but lacked connections with actual asteroid surface data. The next challenge is to use detailed geophysical features returned by spacecraft equipped with state-of-the-art instruments to constrain or even identify the interior properties as well as the associated structural history of visited asteroids. This provides an opportunity to infer evolutionary scenarios from these asteroids' parent-body disruption to their current states along with important insights for future asteroid space exploration. In this study, we tackle this challenge.**”, in the revised manuscript.

Following this new paragraph, we then elaborated on the complex geophysical features of Bennu returned by the OSIRIS-REx mission and added the following statement to highlight the difficulty of linking theoretical/numerical analyses with actual asteroid surface data and to present our methodology, “**Interpretation of this complex set of information requires a comprehensive understanding of asteroid geophysical processes and their proper modelling. Here, to connect all the clues together and infer Bennu's reshaping history in a self-consistent scenario, we performed numerical simulations using the Soft-Sphere Discrete Element Method (SSDEM; Sánchez & Scheeres 2011, Schwartz et al. 2012) to test Bennu's structural response to YORP-induced spinup.**”, starting from Line 66 in the revised manuscript.

At the end of the **Introduction Section**, we added a sentence to highlight the main results and show how the goal of tackling the mentioned challenge is achieved, i.e., “**By exploiting the numerical results and making comparisons with Bennu’s geophysical features, we quantify the interior properties and derive a unified evolutionary scenario for Bennu, which allows us to draw general implications for the structural evolution of top-shaped rubble piles.**”, in the revised manuscript.

Major comment 1: Title: You are mainly talking about Bennu, this paper is not really about two asteroids, but one. Please change your title to reflect this.

Response: We modified the title to “**Using Bennu’s external properties as a guide to interiors and structural histories of top-shape asteroids**” in the revised manuscript.

Major comment 2: In 68: It seems to me you are talking about Reynolds dilation. This phenomenon is common to granular materials, but it depends on the initial condition of the medium. This was also observed by Sanchez and Scheeres (2012). This is, if is all well compacted, the volume will have to increase to allow failure; however, if the initial state of the medium is very loose, it will continually compact under shear. If you only see dilation, it seems to me you always start from a well compacted structure which might not be realistic for an asteroid that is formed under micro-gravity conditions. Did you do some experiments with a more porous structure? Why not?

Response: We agree with the reviewer that the volume change behaviours in granular materials when they are subjected to shear loading depends on their initial packing condition. This is one of the main reasons why we prepared our rubble-pile model carefully by modelling the gravitational collapse of a cloud of particles (see **Methods Section “Bennu rubble-pile model”** for details about our rubble-pile construction procedure). As the accretion and formation process of a rubble-pile body under microgravity conditions is explicitly considered in our study, we believe that our rubble-pile model can well represent the compactness of a real rubble-pile asteroid.

In effect, the initial internal packing efficiency of our rubble-pile model is about 67% (i.e., the macroporosity ~33%). This packing efficiency is within the estimated range of Bennu’s global packing efficiency inferred from its bulk density (1190 kg/m^3), i.e., 50% (assuming the grain density resembling that of CM meteorites, $\sim 2200 \text{ kg/m}^3$) to 75% (assuming the grain density resembling that of CI meteorites, $\sim 1570 \text{ kg/m}^3$; Barnouin et al., 2019). Furthermore, the remote-sensing thermal inertia measurements obtained by the OSIRIS-REx team show that the boulders on Bennu are highly porous, with two main types of microporosity of ~50% and ~30%, which is much larger than the microporosity of the carbonaceous meteorites (Rozitis et al., 2020). This analysis implies that the grain density of Bennu’s materials would be smaller than that of those meteorites, and so, Bennu would have a lower macroporosity and, thus, a higher packing efficiency. In fact, a recent analysis shows that the average of the estimated bulk densities of Ryugu sample particles is $1282 \pm 231 \text{ kg/m}^3$, suggesting a high microporosity down to the millimetre scale (Yada et al., 2021). If the bulk densities of these sample particles provide an unbiased density value for the material on Ryugu, the macroporosity of Ryugu is only ~10% and its packing efficiency is ~90%. Given the similarities of the physical and chemical properties of Bennu and Ryugu, it is very likely that the microporosity of Bennu’s particles is also large. Therefore, the packing structure of our rubble-pile model is valid for representing the structure of Bennu, and it is very unlikely that Bennu would have a lower packing efficiency than that of our model. It is thus unnecessary to consider a more porous structure in this study.

We would also like to kindly point out that the method we used to calculate the internal packing efficiency is essentially different from that of Sánchez and Scheeres (2012). In the study of Sánchez and Scheeres (2012), they used the average filling fraction to evaluate the packing efficiency, where the average filling fraction is obtained from the summation of the exact mass of the particles and the volume of the dynamically equivalent equal-volume ellipsoid of the entire rubble-pile model (see the first paragraph in their Page 881). By this definition, the value of the average filling fraction is largely affected by the overall shape of the rubble pile.

Therefore, the method of Sánchez and Scheeres (2012) in evaluating the packing efficiency cannot eliminate the boundary effect and cannot reveal the internal deformation of the rubble-pile model. It is thus unclear if the decrease in the average filling fraction observed by Sánchez and Scheeres (2012) is due to the surface particle movement or the internal deformation.

In our study, we can explicitly calculate the void space between individual particles in a rubble-pile model by using the Voronoi tessellation analysis (see Section 3.2 and Fig. 4 in Zhang et al. 2017 for details). Therefore, we can eliminate the effect of surface particles and just use the internal particles to evaluate the internal packing efficiency. By this means, the volume change behaviours in the interior of a rubble-pile model during the YORP spin-up can be characterised. Our simulations and analyses provide a novel approach to differentiate the internal structural failure from the surface mass movement, and this is not presented in any of the previous works mentioned by the reviewer.

Major comment 3: In 69-70: This was first observed by Hirabayashi (2015) “Failure modes and conditions of a cohesive, spherical body due to YORP spin-up”.

Response: We disagree with the reviewer on this statement. To help with the discussion, the original statement the reviewer referred to is reproduced as follows, “*For $\varphi > 35^\circ$, some local surface regions near the equator experience outward accelerations before internal slopes exceed φ (Fig. 2b, c). Particles at these regions can be lofted into some close orbits above the surface*”.

The study of Hirabayashi (2015) used an analytical elastic model and a plastic finite element model to explore the failed regions in a spherical body at different spin periods. The two models are based on the continuum medium method, which can neither reveal the acceleration direction and magnitude of individual particles nor infer the movement of surface particles (e.g., landslides and lofting). Therefore, it is incorrect to state that these findings based on our discrete element analyses here were first observed by Hirabayashi (2015).

Major comment 4: In 74-78: There are no details about this, so I will assume that your cohesive aggregates are completely homogeneous. This observation, that the core of a self-gravitating aggregate is going to fail before the surface and that it will need to be stronger to be stable at elevated spin rates was first observed and reported by Hirabayashi, et al, Internal structure of asteroids having surface shedding due to rotational instability (2015). Then this was further studied by Sanchez and Scheeres (2018), Rotational evolution of self-gravitating aggregates with cores of variable strength. These studies certainly worked with spherical and prolate shapes and you have chosen the shape of a real asteroid, but if the observation is exactly the same, you should acknowledge that and point to the fact that it seems to be a more general outcome than previously thought.

Response: The structure discussed here is homogeneous. To clarify, we added a sentence, “**Taking the homogeneous Bennu-shaped model as an example,**”, in Line 100 in the revised manuscript.

There are fundamental differences between our findings and the previous findings on the failure patterns of rubble-pile asteroids. As pointed out by the reviewer, the main findings of Hirabayashi et al. (2015) and Sánchez and Scheeres (2018) are that, “*the core of a self-gravitating aggregate is going to fail before the surface and that it will need to be stronger to be stable at elevated spin rates.*”, i.e., in homogeneous and weak-core aggregates, the internal core would always fail before the surface shell.

However, this is not the case as shown by our study. As summarised in Supplementary Table 1, our results show that a rubble-pile model with homogeneous cohesion can also fail superficially when the friction angle is larger than 32 degrees. The internal failure mode (i.e., the Type II failure mode) only occurs for some combinations of friction and cohesion values (the original text the reviewer mentioned corresponds to our simulation results with a friction angle of 29 degrees). This high sensitivity of the failure mode to the material

properties revealed by the current study lays out the foundation for interpreting the interiors and structural histories of a rubble-pile asteroid such as Bennu. These quantitative relations between the failure mode and the material properties revealed by our study have never been reported in Hirabayashi et al. (2015), Sánchez and Scheeres (2018), and any previous studies.

Below, we explain this discrepancy between our findings and those in Hirabayashi et al. (2015) and Sánchez and Scheeres (2018) based on the two methods they applied in their studies:

⓪ In the continuum-theory analyses of Hirabayashi et al. (2015), the friction angle is assumed to be 35 degrees, and the failure mode they predicted (i.e., “*always fail internally before it fails superficially*” as described in their conclusion) is clearly opposite to our prediction (i.e., the surface landslide and mass-shedding failure mode observed in the case of the homogeneous structure; see Supplementary Table 1). The inconsistency between our numerical results and the continuum-theory analyses of Hirabayashi et al. (2015) mainly comes from the differences between the friction and cohesion mechanism in the discrete-element modelling and that in the continuum method. The shear and cohesive strength can be completely controlled by two specific variables in the continuum method (e.g., μ and C in the Drucker–Prager failure criterion; Eq. (4)), while, in an actual rubble pile, the shear and cohesive strength would depend on the local packing, which can only be captured by a discrete approach. As shown in Zhang et al. (2017), the local packing efficiency and coordination number for the surface particles is much lower than that for the interior particles. Therefore, surface shedding can still occur for a rubble-pile object with some combinations of friction and cohesion values as revealed by our study.

⓪ In the SSDEM analyses of Hirabayashi et al. (2015) and Sánchez and Scheeres (2018), the friction angle is also set to be 35 degrees, and they found that the homogeneous rubble pile deformed internally when the body started to fail (i.e., Figure 5 in Sánchez and Scheeres (2018)), which is also clearly different than what we found in our study. There are two main reasons causing this difference:

- A) Resolution and shape of rubble-pile models: the spherical low-resolution model ($N \sim 3,000$) used in Hirabayashi et al. (2015) and Sánchez and Scheeres (2018) does not reflect the granular rough surface of an actual asteroid, while our high-resolution model ($N \sim 41,000$) can characterise the geophysical features (such as the shape and surface slopes) of such asteroids. The mobility of surface material can be captured by the increased roughness of the surface, which can lead to the mass-shedding failure mode we observed in our study.
- B) Spin-up procedure: in the SSDEM studies by Sánchez and Scheeres, their rubble-pile models were spun up by adding angular velocity to the aggregate and the constituent particles by discrete increments. The resulting Euler acceleration is much larger than the centrifugal acceleration, which makes their failure behaviour prediction inapplicable to testing the effect of YORP spinup. For example, the spin rate was increased by 4.5×10^{-6} rad/s every 3000 s in Sánchez and Scheeres (2016). Given the time step they used is about 0.25 s (based on the indication in Sánchez and Scheeres, 2012; as the 2016 and 2018 papers do not have information about the timestep setup), the resulting spin-up rate is about 1.8×10^{-5} rad/s² and the corresponding Euler acceleration at the moment of changing the spin rate for a given particle is $1.8 \times 10^{-5} r_p$ m/s², where r_p is the distance (unit: m) of the particle to the rubble pile’s spin axis. Compared to the centrifugal acceleration at Bennu’s current spin period (4.296 hr), i.e., $1.7 \times 10^{-7} r_p$ m/s², the Euler acceleration is two orders of magnitude larger. The movement of the constituent particles right after the moment of changing the spin rate are mainly affected by the Euler acceleration rather than the centrifugal acceleration, and the failure behaviours are results of both accelerations. However, in reality, the Euler acceleration of the YORP effect is extremely small (e.g., $8.5 \times 10^{-18} r_p$ m/s² for Bennu) and should not have influence on the failure behaviours. Therefore, the discrete spin-up procedure used in the SSDEM studies of Sánchez and Scheeres is unreliable for predicting the failure behaviours of an SSDEM rubble-pile model under the YORP spin-up effect. In

the current study, we designed a quasi-static spinup path whose Euler acceleration is about $1.9 \times 10^{-9} \text{ m/s}^2$, which is two orders of magnitude smaller than the centrifugal acceleration, and thus, the failure behaviour of a rubble-pile body under YORP spinup can be readily revealed by the modelling.

In conclusion, the quantitative relations between the failure modes and the material properties revealed by our study are novel and different from the findings in previous studies. To stress the advances of our current study, we added the following discussions, “**The dependency of failure modes on asteroids’ material properties revealed in the current study is much more complex and sensitive than previous studies predicted. It has been suggested, based on both continuum-theory analyses (Hirabayashi et al. 2015) and SSDEM simulations (Sánchez and Scheeres 2018), that in homogeneous spherical rubble piles, the internal core would always fail before the surface shell. However, our results show that a rubble-pile model with homogeneous structures can also fail superficially via the Type III mass-shedding mode when the friction angle is sufficiently large. The internal failure mode (i.e., the Type II failure mode) only occurs for some combinations of friction and cohesion values. This discrepancy could be mainly due to the utilisation of high-resolution models and the quasi-static spinup procedure in the current study that can characterise the geophysical features (such as the shape and surface slopes) and YORP-induced structural response of an actual asteroid. Both cannot be captured by the static continuum analyses and the low-resolution SSDEM models used in these previous studies. This high sensitivity of the failure mode to the material properties revealed here lays out the foundation for interpreting the interiors and structural histories of a rubble-pile asteroid such as Bennu.**”, in Line 146–159 in the revised manuscript.

***Major comment 5:* In 79: You have not discovered these patterns in the general sense, you have observed them for a particular shape as they were already discovered by the previously cited authors.**

Response: To our best knowledge, the Type I landslide and Type IV tensile disruption failure patterns shown in Figure 1 and Figure 2 have never been reported by any of previous theoretical or numerical studies including the ones mentioned by the reviewer. It is apparent that none of previous SSDEM studies could have sufficient rubble-pile model resolution to carry out such analyses. The Type II internal deformation and Type III mass shedding failure behaviours were first reported via HSDEM numerical simulations by three of our co-authors (Walsh et al., 2008), and, in the current study, we design the stress/slope analysis approaches to reveal the stress/slope distribution patterns behind the observed failure behaviours (see Figure 2). These stress/slope distribution patterns have not been shown in any of these previous studies.

To be more specific, we modified the original text to, “According to the discovered instability patterns **and stress/slope distribution characteristics** of each failure mode, ...”, in Line 118 in the revised manuscript.

***Major comment 6:* In 100-104: Hirabayashi, Sanchez and Scheeres have worked on this specific topic for years, I find it very strange that most of their papers are not cited. The inclusion of strong and weak cores was the subject of some of the papers I have cited above. You would do well in getting acquainted with their results.**

Response: Since the current paper is a research paper not a review paper and there is a strict limit on the number of references (i.e., 50 for the Letter style we used to prepare our original manuscript), we can only cite the papers that are very relevant for our current study (e.g., the observational results of the OSIRIS-REx mission). Nevertheless, in our original manuscript, we have cited 5 papers led by the three researchers the reviewer mentioned, i.e., Scheeres et al. (2010, 2020), Sánchez & Scheeres (2016), Hirabayashi (2015), Hirabayashi et al. (2020), to acknowledge their contributions in this field and facilitate the relevant discussions, and some related results of other papers mentioned by the reviewer are also discussed in the above studies.

As our manuscript are reshaped into an Article style and we are allowed to have up to 70 references, we agree with the reviewer that it is beneficial to discuss and make comparisons with the main results of Hirabayashi et al. (2015) and Sánchez and Scheeres (2018). Please see our response to Major comment 4 for the detailed revisions.

Major comment 7: In 123-124: Or it could be that, though maybe not as likely, as proposed by Tardivel, et al (2018), “Equatorial cavities on asteroids, an evidence of fission events”, a cohesive chunk of material simply was detached from the equatorial region. Have you completely discarded this possibility? Why?

Response: The fission scenario cannot systematically account for the observed morphologies of the equatorial cavities (see Figure A) as well as its other geophysical characteristics on Bennu for the following reasons:

(1) This scenario is problematic to produce the bowl-shaped cavities. These large equatorial cavities have bowl-like shapes with diameters of ~100 m according to the high-resolution (~1 m) Digital Terrain Model of Bennu. If the centrifugal forces were sufficient to lift a boulder of that size off the surface, surely it would also push surrounding and underlying material out to fill the cavity. The resulting morphology would be very different from what was observed on Bennu.

(2) This scenario is inconsistent with the surface mass movement activities observed on Bennu. Since the surface cohesion is minimal on Bennu as inferred from the surface mass movement (Figure 1), it is implausible that a cohesive chunk of material with sizes of ~100 m can be detached from the equatorial region but the cohesionless surface material can still be bounded and remained on Bennu.

Furthermore, decades of numerical, experimental, and theoretical works have shown that impacts produce bowl-shaped cavities, which can consistently explain the morphology of these equatorial cavities. Therefore, the fission scenario is considered to be inapplicable for the case of Bennu.

[redacted]

Major comment 8: In 141-142: Again, this model was first explored, proposed and implemented by Sanchez and Scheeres (2014), The strength of regolith and rubble pile asteroids. However, it has not been cited.

Response: The heterogeneous model we proposed here is totally different from the model proposed by Sánchez and Scheeres (2014). As presented in Figure 4a, we consider large cohesive granular aggregates with radii of ~100 m embedded in cohesionless granular medium, while the model of Sánchez and Scheeres (2014) considers two big particles connected by small cohesive particles that are pulled apart. The two models are reproduced in Figure B for comparison. It is obvious to see that these are very different models and very different scenarios.

[redacted]

Major comment 9: In 152-154: This could be true; however, it does not explain its symmetry. Can you explain the symmetric, "squarish equatorial shape?"

Response: As shown in Figure 4d, the internal deformation process is symmetry along the $x = 0$ and $y = 0$ axis. For clarification, we modified the description to, “the heterogeneous landslides facilitate the formation of a non-circular equator (Fig. 4e), and the accompanying internal deformation, which is symmetric along the $x = 0$ and $y = 0$ axis (Fig. 4c, d), leads to a squarish equatorial shape (Fig. 4b), resembling the shape of Benuu (Barnouin et al. 2019)”, in Line 222–225 in the revised manuscript.

Major comment 10: In 158-159: Again, the core could indeed be more porous or weaker than the surface, but this cannot be readily linked to the internal shear deformation. As explained above, this decrease in filling fraction happens only when the initial aggregate is well packed. This is the starting point you have chosen for your simulations, but you have not justified why this would be a realistic starting point for Benuu. Furthermore, if this is an unlikely starting point, I think most of what you have observed in your simulations could be inapplicable to Benuu.

Response: Please see our response to Major comment 2 for the reasons why our rubble-pile model is justified for the current study.

Major comment 11: In 161-162: What do you mean by your approach? Are you talking about using SSDEM simulations to form self-gravitating aggregates with different material and bulk characteristics to then spin them up and associate their findings to observed features in small bodies? Scientist have been doing numerical simulations using soft-sphere DEM codes since 2008 when Sanchez and Scheeres introduced them to Planetary Sciences. Richardson, Michel and Schwartz did this using a HSDEM code.

Hirabayashi, Rozitis and Holsapple have been doing theoretical and numerical work for the last 2 decades. Please ensure to cite all relevant literature and consider summarizing previous contributions to the field.

Response: Scientists have been effectively doing theoretical studies and numerical simulations to understand the evolution of asteroids during YORP spinup for a long time. However, this is the first time that a study like ours is based on the geophysical features characterised in great detail by spacecraft on an asteroid. Therefore, here, we go beyond the pure theoretical and numerical studies and make consistent and systematic connections to real asteroid data. For clarification, we modified the sentence to, “**By linking observed geophysical features with numerical experiments, we probe Bennu’s interior and identify associated paths in its structural evolution.**”, in Line 234–235 in the revised manuscript.

Major comment 12: In 165-166: This was expressed in Sanchez and Scheeres (2014) and expressly used in their 2018 paper about cores of variable strength.

Response: The study of Sánchez and Scheeres (2014) is cited in the place indicated by the reviewer (Line 244) to support the idea of the presence of local concentrations of fine grains.

Major comment 13: In 169-170: This statement makes no senses unless you cite Sanchez and Scheeres (2014) about the strength of rubble pile asteroids.

Response: The study of Sánchez and Scheeres (2014) is cited in the place indicated by the reviewer (Line 249) to explain the effect of fine grains.

Major comment 14: fig 4: Compare to the images in Hirabayashi, et al, Internal structure of asteroids having surface shedding due to rotational instability (2015), or Sanchez and Scheeres (2018), Rotational evolution of self-gravitating aggregates with cores of variable strength. Anthem, they explain how the strength of the core is directly linked to the failure mode of the internal structure. Please, cite those earlier studies, discuss potential overlaps and stress how your study advances those.

Response: The two studies were cited and discussed in the revised manuscript. Please see our responses to General comment and Major comment 4 for details.

Major comment 15: In 321: The Soft-Sphere method was first implemented for the simulation of small bodies by Sanchez and Scheeres, the numerical method itself was first implemented for granular materials by Cundall and Strack. The authors seem to imply here that this method is their invention, which it is most certainly not. Please refer to those relevant literature.

Response: The original text the reviewer referred to is, “*We used the high-efficiency parallel N-body code, pkdgrav (Richardson et al., 2000; Stadel, 2001), and its soft-sphere discrete element modelling (SSDEM) framework (Schwartz et al., 2012), including an improved rolling friction (Zhang et al., 2017) and cohesion (Zhang et al., 2018) model, to solve the gravity and contact interactions between spherical particles representing the components of a simulated rubble pile.*” This is an even-handed statement, which presents the facts about our numerical method. We do not see how this statement can be interpreted as that we invent the soft-sphere method.

On the contrary, we cited the paper of Schwartz et al. (2012) to refer the readers for more details if they want to know about the implementation of the soft-sphere method in pkdgrav. A brief review on the soft-sphere method and its application in the field of small bodies can be found in the introduction of Schwartz et al. (2012). The two studies mentioned by the reviewer, i.e., Cundall & Strack (1979) and Sánchez & Scheeres (2011), have been cited and discussed in this model implementation paper (i.e., Schwartz et al., 2012).

For clarification, in the revised manuscript, we also cite the paper of Sánchez & Scheeres (2011) when we first introduced the SSDEM in Line 69 to show that the SSDEM have been implemented in different codes by different researchers.

Major comment 16: In 354: This figure does not really contain anything to help me see that the process is quasi-static. Please, add another figure.

Response: We are not sure if the reviewer looked at the correct figure. By definition, a quasi-static process (also known as a quasi-equilibrium process) is a dynamical process that happens slowly enough for the system to remain in internal dynamical equilibrium. As shown in Supplementary Figure 1, during the spin-up process, the axis ratio and internal packing efficiency remain constant at slow spin states, and these two variables gradually reduce when the spin rate is close to their spin limit. After the external loading is removed, these two variables quickly achieve a new equilibrium state. All the above observations indicate that the process we modelled is quasi-static.

Major comment 17: In 359-361: Structural failure, though it is linked to body deformation, is directly detected by a yield criterion. This has been extensively studied in mechanical and civil engineering. Its application to asteroids has been carried out mainly by Sharma, Holsapple and Hirabayashi for the last two decades to mention a few. Please, use that; you have the tools to calculate the stress tensor which is what you need as input for the yield criterion calculation.

Response: In the case of internal deformation, it is true that the stress yield criterion would give a more accurate estimate on the critical spin limit. In our previous study, i.e., Zhang et al. (2017), we have developed the tools to use this yield criterion and made comparisons with the structural failure criterion based on changes in shape, i.e., the criterion used in the current study. We found that the critical spin period obtained by the latter criterion is slightly shorter than that obtained by the former criterion. The reason is that the structural failure behaviours, such as body deformation and mass shedding, lags behind the structural yielding. Nevertheless, the detected failure behaviours are not affected by the choice of the criterion.

However, in the case of surface mass shedding failure mode, the internal structure would not yield, and therefore, this yield criterion cannot be applied. This is the reason why we need to use the surface slope/cohesion analyses to assess the failure condition of the rubble pile.

In the current study, in order to make a systematic comparison between different failure modes, we can only use the failure criterion that can be observed in all the failure modes. This is the reason why we need to use the shape change criterion rather than the yield criterion.

Major comment 18: In 370-373: I think it is Ok to cite preliminary work and results, but I don't think it is appropriate to say that these results fully justify to do something without any caveats. If these results are preliminary, it means that a complete study is not yet finished and your conclusions could change. Please, either remove or change this statement to reflect this.

Response: To justify the continuous spinup path used in our simulations, we added the following sentences to discuss the effect of the two scenarios, “**If this is the case, Bennu would be spun up in a relatively constant rate; otherwise, due to the randomness of the YORP torque, the spin-rate-doubling timescale would be longer than the current YORP torque predicts. Nonetheless, the structural failure behaviours of Bennu at its spin limit should be independent of the YORP torque magnitude.**”, in Line 379–383 in the revised manuscript.

Major comment 19: In 396-398: This analysis is similar to that carried out by Patel and Hartzell (2021), "A Model to Predict the Size of Regolith Clumps on Planetary Bodies." Please, see if a citation is needed

here or if you need to modify this section to use their work. If not, please explain why this is better, or more applicable, than what they had done.

Response: The analysis carried out by Patel & Hartzell (2021) is not similar to the analysis presented in the current study. The model of Patel & Hartzell (2021) is developed to predict the size of regolith clumps that can form under different gravity environments, while the model presented in the current study is developed to predict the dynamical surface cohesion as a function of surface slope for a grain with a given size (see Figure 2d for an example). Nonetheless, we added a sentence to acknowledge the possible formation of clumps on asteroids, i.e., “As $\sigma_p^{\text{sur}} \propto \sigma_p$, smaller particles are generally more difficult to loft. These macroscopic particles could be individual boulders or agglomerates of smaller grains that clump together due to physical interlocking and/or chemical attractions. The formation of cohesive clumps in surface mass wasting is commonly observed terrestrially (Sánchez et al. 2021), and theoretical analyses have shown that clumps of cm-scale and smaller grains are possible to form on asteroids and may be easier to detach from a surface than their constituent grains (Patel & Hartzell 2021).”, in Line 408–414 (following the original text the reviewer referred to) in the revised manuscript.

Major comment 20: In 442: This is the kinetic angle of repose.

Response: By definition, the kinetic angle of repose is measured in a dynamical process where sediment grains are moving continuously down an inclined plane, while the static angle of repose is measured at the point where the slope instability is initiated. Therefore, it is clear to see that the angle of repose we measure here is the static angle of repose rather than the kinetic angle of repose.

Major comment 21: In 443-445: Yes, theoretically these two should be equal; this is a known result.

Response: In fact, the angle of repose and the internal angle of friction of a granular material depend on the material properties as well as the levels of confinement and gravity. It is not always true that the angle of repose is equal to the internal friction angle because the behaviours of granular materials under low confining pressure is considerably different than that under zero confining pressure, as found in a number of previous studies (please see a relatively recent review paper by Al-Hashemi & Al-Amoudi, 2018, on this subject). Therefore, our finding here reveals that the internal friction angle is very close to the angle of repose under the extremely low-gravity environment of Bennu.

Minor comment 1: In 50: Please, provide the particle size range.

Response: We added the description, “... particles with radii ranging from ~4 m to ~12 m following a power-law ...”, to indicate the particle size ranges in Line 72 in the revised manuscript.

Minor comment 2: In 121: “... Bennu’s surface is due to...” -> “... Bennu’s surface could be due to...”

Response: “... Bennu’s surface is due to ...” is replaced by “... Bennu’s surface is more likely due to ...” in Line 190 in the revised manuscript.

Minor comment 3: In 335: It seems to me that you are using a linear spring-dashpot model here. If you are, please be explicit about it as non-linear models could also be used.

Response: We added a sentence, “Briefly, the SSDEM model includes a linear spring-dashpot normal contact force, a normal cohesive force, a spring-dashpot-slider tangential contact force, and two spring-dashpot-slider rotational torques in the rolling and twisting directions.”, to describe the SSDEM model details in Line 314–316 in the revised manuscript.

Minor comment 4: In 396-397: “... a initially...” -> “... an initially...”

Response: “... a initially ...” is replaced by “... an initially ...” near Line 405 in the revised manuscript.

Final comment: In this report, I have tried to highlight obvious omissions to previous work to make it obvious that the paper does not fulfill the criterion for publication of Nature Communications. This is not something that can be solved just by adding some citations as, even if added, this will simply highlight the fact that the paper is an application, albeit an interesting one, of previously known results. As such, it is my belief that the paper does not “represent an advance in understanding likely to influence thinking in the field.” To show this in a simple detail, the following conclusion can be found in Sanchez and Scheeres (2018), “Rotational evolution of self-gravitating aggregates with cores of variable strength”: “This implies that the specific shapes, spin rates and surface features that have been observed in asteroids are an expression of their hidden interiors and, as such, should provide insight about the formation and evolution processes of these bodies”, which is the main idea of the title and the paper. Yet, this paper is never cited. Apart from this, the authors do not justify their choice of initial packing and this carries many implications for the subsequent evolution. How this affects the results is not a trivial matter, but the authors ignore this and simply apply all their results to Bennu with no caveats at all. In fact, a change in the initial packing of the body could lead to a completely different evolution.

Response: We thank the reviewer for the above comments, which helped us to improve the manuscript and highlight the major contributions of the current study more clearly. As discussed in our responses to all the comments above (especially to General comment and Major comment 4), the current study represents an important advance in understanding asteroid interiors and their structural evolutions. Regarding the justification of our choice of initial packing, our response to Major comment 2 has addressed this issue and demonstrated that the packing structure of our rubble-pile model is valid for representing the structure of Bennu. Furthermore, the semi-analytical method developed in the current study does not rely on a particular initial packing of the considered body, and the failure modes (Figure 3) derived by this semi-analytical method are consistent with our numerical results, which indicates that the evolutionary scenarios and their dependency on the body’s surface and interior properties revealed by the current study are robust.

Once again, thank you very much for your insightful comments and suggestions.

References

- Al-Hashemi, H. M. B., & Al-Amoudi, O. S. B. (2018). A review on the angle of repose of granular materials. Powder technology, 330, 397-417.
- Barnouin, O. S., Daly, M. G., Palmer, E. E., ... & Lauretta, D. S. (2019). Shape of (101955) Bennu indicative of a rubble pile with internal stiffness. Nature geoscience, 12(4), 247-252.
- Cundall, P. A., & Strack, O. D. (1979). A discrete numerical model for granular assemblies. geotechnique, 29(1), 47-65.
- Rozitis, B., Ryan, A. J., Emery, J. P., ... & Lauretta, D. S. (2020). Asteroid (101955) Bennu’s weak boulders and thermally anomalous equator. Science Advances, 6(41), eabc3699.
- Hirabayashi, M. (2015). Failure modes and conditions of a cohesive, spherical body due to YORP spin-up. Monthly Notices of the Royal Astronomical Society, 454(2), 2249-2257.
- Hirabayashi, M., Nakano, R., Tatsumi, E., Walsh, K. J., Barnouin, O. S., Michel, P., ... & Lauretta, D. S. (2020). Spin-driven evolution of asteroids' top-shapes at fast and slow spins seen from (101955) Bennu and (162173) Ryugu. Icarus, 352, 113946.

- Patel, A., & Hartzell, C. (2021). A Model to Predict the Size of Regolith Clumps on Planetary Bodies. *The Planetary Science Journal*, 2(5), 196.
- Sánchez, P., & Scheeres, D. J. (2011). Simulating asteroid rubble piles with a self-gravitating soft-sphere distinct element method model. *The Astrophysical Journal*, 727(2), 120.
- Sánchez, P., & Scheeres, D. J. (2016). Disruption patterns of rotating self-gravitating aggregates: a survey on angle of friction and tensile strength. *Icarus*, 271, 453-471.
- Scheeres, D. J., Hartzell, C. M., Sánchez, P., & Swift, M. (2010). Scaling forces to asteroid surfaces: The role of cohesion. *Icarus*, 210(2), 968-984.
- Scheeres, D. J., French, A. S., Tricarico, P., Chesley, S. R., Takahashi, Y., Farnocchia, D., ... & Lauretta, D. S. (2020). Heterogeneous mass distribution of the rubble-pile asteroid (101955) Bennu. *Science advances*, 6(41), eabc3350.
- Schwartz, S. R., Richardson, D. C., & Michel, P. (2012). An implementation of the soft-sphere discrete element method in a high-performance parallel gravity tree-code. *Granular Matter*, 14(3), 363-380.
- Walsh, K. J., Richardson, D. C., & Michel, P. (2008). Rotational breakup as the origin of small binary asteroids. *Nature*, 454(7201), 188-191.
- Yada, T., Abe, M., Okada, T., ... & Tsuda, Y. (2021). Preliminary analysis of the Hayabusa2 samples returned from C-type asteroid Ryugu. *Nature Astronomy*, <https://doi.org/10.1038/s41550-021-01550-6>.
- Zhang, Y., Richardson, D. C., Barnouin, O. S., Maurel, C., Michel, P., Schwartz, S. R., ... & Li, J. (2017). Creep stability of the proposed AIDA mission target 65803 Didymos: I. Discrete cohesionless granular physics model. *Icarus*, 294, 98-123.
- Zhang, Y., Richardson, D. C., Barnouin, O. S., Michel, P., Schwartz, S. R., & Ballouz, R. L. (2018). Rotational failure of rubble-pile bodies: influences of shear and cohesive strengths. *The Astrophysical Journal*, 857(1), 15.

REVIEWER COMMENTS

Reviewer #1 (Remarks to the Author):

=====

Overall, the authors have addressed my main comments to a satisfactory level. One of my main points was that the authors needed to cite relevant work in this area, and also to demonstrate how their work advances the state-of-the-art. Regarding the latter, while the authors did not address this comment in their response to my report, they actually provide all the technical detail I would like to have seen in the response to Reviewer 3 (see pages 6-8). In fact, this material could potentially be added to the Methods section itself, but that can be decided by the authors, and/or the editor. I recommend publication in more or less its current form.

Some additional suggested minor comments are included below:

Page 2, line 29: Why the cut-off at 10km for rubble piles?

Page 2, Line 43: In my opinion “asteroid’s parent body” is misapplied here. The term ‘parent body’ implies a larger asteroid being divided up into many small pieces, and Bennu may not have been produced from such an event.

Page 3, Line 65: Why do you say that Bennu is “under dense”?

Page 6, Line 152-153: Please consider rewording the following phrase:

“This discrepancy could be mainly due to the utilisation of high-resolution models”

to

“This discrepancy could be mainly due to the utilisation of increased-resolution SSDEM models”.

Then further down change,

“low-resolution” to “lower-resolution”.

The models will always increase in resolution with time, so best to avoid absolute terms like “high” and “low”, in this context at least.

Page 9, bottom paragraph: Could YORP-induced “seismic shaking” contribute too, producing a ‘Brazil-Nut Effect’?

Figure 3: Please add some more description of what the yellow bands represent and how they are produced (in the main text or caption).

=====

Reviewer #2 (Remarks to the Author):

I have gone through the comments from the 3 reviewers and the responses by the authors.

To follow the reviewers' suggestions, the article has been largely extended. As far as I understand from one of the authors' comments, the style type has been changed in order to allow a longer format and more references. The submission was in a Letter style, and now it is the Article style.

I don't know how this affects the revision process.

In any case, I consider this extension to be welcome as it greatly improved the manuscript.

In view of the responses given by the authors to the comments of the 3 reviewers, and in particular to the criticism by Rev.#3, my opinion is that the authors had successfully addressed the most relevant comments.

My original comments were already very positive, and I keep this evaluation.

Reviewer #3 (Remarks to the Author):

I would like to thank the authors for their thorough reply to my comments and for revising their paper. They have now made clear how their work is indeed an advancement and not only an application of previously acquired knowledge.

I have nothing else to add.

I am happy to recommend this paper for publication without any further modification. Well done!

Response Letter to Reviewers

We thank the reviewers for their careful reading and insightful comments. We are glad to know that our previous responses were well received by the reviewers.

Below, we address the new comments from Reviewer #1 (presented **in bold**) point by point and highlight the corrections in the manuscript **in red colour**. All the changes are also highlighted **in red colour** in the revised manuscript.

Reviewer #1:

General comment: Overall, the authors have addressed my main comments to a satisfactory level. One of my main points was that the authors needed to cite relevant work in this area, and also to demonstrate how their work advances the state-of-the-art. Regarding the latter, while the authors did not address this comment in their response to my report, they actually provide all the technical detail I would like to have seen in the response to Reviewer 3 (see pages 6-8). In fact, this material could potentially be added to the Methods section itself, but that can be decided by the authors, and/or the editor. I recommend publication in more or less its current form.

Response: We thank the reviewer for the suggestion. In the revised manuscript, we added a paragraph at the end of the Methods Section “Bennu rubble-pile model”, i.e., “**Note that, due to computational constraints, previous SSDEM simulations dedicated to the study of rubble-pile failure behaviour, e.g., Hirabayashi et al. 2015 and Sánchez & Scheeres 2018, commonly used rubble-pile models that consisted of $N \sim 3,000$ spheres with size differences $R_{\max}/R_{\min} < 1.5$, which cannot capture the high-accuracy shape model of a space-mission target like Bennu, and thus limit the modeling of the target’s geophysical features and evolution. In this study, thanks to the hierarchical tree data structure and high-efficiency parallelization of our modelling code, *pkdgrav* (Richardson et al. 2000; Stadel 2001; see below), a self-gravity N -body system with N up to 10^5 can be readily modelled within acceptable computational costs (Zhang et al. 2021). By using the increased-resolution models with relatively large particle size differences, we are able to mimic the observed boulder distribution on Bennu and capture the shape of Bennu at relatively high resolution. This increased-resolution model also enables us to design the novel analytical approach (see Methods Sections “Dynamical internal slope and cohesion distribution of a rubble pile” and “Dynamical surface slope and cohesion distribution of a rubble pile”) to calculate the internal stress-state and surface slope/cohesion distribution (see Fig. 2 for examples), which helps reveal the underlying mechanisms that lead to different failure behaviour and provide new insights into the surface and internal properties of top-shaped asteroids and their geophysical evolution. As demonstrated in Supplementary Fig. 4, the surface slope distribution of our rubble-pile models resembles that of Bennu, which allows us to carry out the surface mass movement analyses and make direct comparisons with the geophysical features discovered by the OSIRIS-REx mission (see Fig. 1). These modelling and data analysis techniques in turn enable us to constrain Bennu’s surface and interior material properties and develop a unified evolutionary scenario for Bennu as discussed in the main text.”, to state the technical detail about how the present work advances the state-of-the-art.**

Minor comment 1: Page 2, line 29: Why the cut-off at 10 km for rubble piles?

Response: This value of 10 km is not meant to be a cut-off, but rather an order of magnitude of the size of the objects that are considered to have rubble-pile structures according to the current understanding of asteroid formation and collisional evolution. The detailed analyses and justification for this size range are discussed in the review paper, Walsh, K.J. *Rubble pile asteroids. Annu. Rev. Astron. Astrophys.* 56, 593-624 (2018). In the

manuscript, this paper is referenced at the end of this sentence in Line 29 to justify this statement. To clarify, we also rephased this sentence to, “Rubble-pile structures are expected to predominate among asteroids with diameters of 200 m to **tens of kilometers according to the current understanding of asteroid formation and collisional evolution** (Walsh 2018)”, in the revised manuscript.

Minor comment 2: Page 2, Line 43: In my opinion “asteroid’s parent body” is misapplied here. The term ‘parent body’ implies a larger asteroid being divided up into many small pieces, and Bennu may not have been produced from such an event.

Response: To make the description more general, we replaced “asteroids’ parent-body disruption” with “**asteroids’ origin**” in Line 44 in the revised manuscript.

Minor comment 3: Page 3, Line 65: Why do you say that Bennu is “under dense”?

Response: We intended to state that Bennu is likely to have a lower-density interior here. This density distribution feature is obtained from Bennu’s gravity field measurement by the OSIRIS-REx mission, as discussed in *Scheeres et al. Heterogeneous mass distribution of the rubble-pile asteroid (101955) Bennu. Science Advance 6, eabc3350 (2020)*. In the manuscript, this paper is referenced in Line 65 to provide the justification. To make it clear, the term “an under-dense interior” is replaced by “**a lower-density interior**” in Line 65 in the revised manuscript.

Minor comment 4: Page 6, Line 152-153: Please consider rewording the following phrase: “This discrepancy could be mainly due to the utilisation of high-resolution models” to “This discrepancy could be mainly due to the utilisation of increased-resolution SSDEM models”. Then further down change, “low-resolution” to “lower-resolution”. The models will always increase in resolution with time, so best to avoid absolute terms like “high” and “low”, in this context at least.

Response: The term “high-resolution” and “low-resolution” in Line 153 and Line 156 were replaced by “**increased-resolution**” and “**lower-resolution**”, respectively, in the revised manuscript.

Minor comment 5: Page 9, bottom paragraph: Could YORP-induced “seismic shaking” contribute too, producing a ‘Brazil-Nut Effect’?

Response: There is no seismic shaking process that could be induced by YORP to the best of our knowledge. The seismic shaking effect could contribute to the percolation of fine particles and the expose of large particles on the surface, although the effectiveness of seismic shaking process in microgravity environments and porous medium is still unclear currently. To acknowledge the possible effect of seismic shaking, we modified the sentence in Line 254 to, “Another possibility is that fines percolate into the subsurface thanks to the structural macroporosity **and/or seismic shaking effects (Tancredi et al. 2012; Perera et al. 2016)**”, in the revised manuscript.

Minor comment 6: Figure 3: Please add some more description of what the yellow bands represent and how they are produced (in the main text or caption).

Response: The yellow bands denote the 1-sigma and 2-sigma distribution of surface slope and cohesion. In the caption, we modified the description to, “The distribution of surface slopes and cohesion of the Bennu-shaped rubble pile are shown as the median (i.e., the dashed yellow curves) and 1-sigma to 2-sigma ranges (**i.e., the yellowish regions with different opacity as indicated by the double-sided arrows**)”, to clarify the meaning of the yellow bands.

The detailed procedure to produce these curves and bands shown in Figure 3 are given in the Methods Section “Theoretical failure conditions”. To clarify, in the revised manuscript, we added, “(see Methods Section “Theoretical failure conditions” for the procedure to generate this diagram)”, in the caption of Figure 3.

Once again, thank you very much for your comments and suggestions.

REVIEWERS' COMMENTS

Reviewer #1 (Remarks to the Author):

Thank you for the revisions and responses to my comments, which have all been fully addressed. I pass on my congratulations to the authors for this study.